# Complexity and weak integration promote the diversity of reef fish oral jaws

M. D. Burns[1,2] ✉, D. R. Satterfield [2], N. Peoples [2], H. Chan [3], A. J. Barley[4], M. L. Yuan [2], A. S. Roberts-Hugghis[2,5,6], K. T. Russell[2], M. Hess[2], S. L. Williamson[2], K. A. Corn[7,8], M. Mihalitsis [2], D. K. Wainwright[9] & P. C. Wainwright [2]

Major trade-offs often manifest as axes of diversity in organismal functional systems. Overarching trade-offs may result in high trait integration and restrict the trajectory of diversification to be along a single axis. Here, we explore the diversification of the feeding mechanism in coral reef fishes to establish the role of trade-offs and complexity in a spectacular ecological radiation. We show that the primary axis of variation in the measured musculo-skeletal traits is aligned with a trade-off between mobility and force transmission, spanning species that capture prey with suction and those that bite attached prey. We found weak or no covariation between about half the traits, reflecting deviations from the trade-off axis. The dramatic trophic range found among reef fishes occurs along the primary trade-off axis, with numerous departures that use a mosaic of trait combinations to adapt the feeding mechanism to diverse challenges. We suggest that morphological evolution both along and independent of a major axis of variation is a widespread mechanism of diversification in complex systems where a global trade-off shapes major patterns of diversity. Significant additional diversity emerges as systems use weak integration and complexity to assemble functional units with many trait combinations that meet varying ecological demands.

The diversification of complex biomechanical and physiological systems is often shaped by important trade-offs[1–4] where changes to a component improve performance in one function but necessarily decrease performance in a second function. Major trade-offs may be substantial organizing forces in evolution, ultimately manifesting as overarching axes of diversification, such as is seen in the wings of birds[5], the legs of frogs[6], the carapace of turtles[7], and the fins of fishes[8]. Trade-offs can have substantial macroevolutionary consequences as they can constrain evolution by limiting the compatibility of certain trait values[2,3,9–11] and the resulting high trait covariation may promote phenotypic macroevolution along an adaptive line of least resistance[10,11]. In this way, trade-offs may induce such strong trait covariations as to limit phenotypes independent of the major trade-off[10,11].

Dominant trade-offs have established primary axes of diversity in many functional systems, yet for many major radiations, there is extensive ecological and morphological diversity that extends well beyond the trend of the primary traits involved in the trade-off. Here, the increasing number and

diversity of traits contributing to performance allows for expanded diversification as unique trait combinations can satisfy different functional demands[12,13]. This may manifest as weakened covariation among traits and expand the range of ecomorphological diversity observed, independent of the major trade-off, by allowing traits to respond more independently to natural selection[14–18]. On macroevolutionary timescales, increased independence of traits within complex mechanical systems may result in a variety of evolutionary pathways not limited by the dominant trade-off axis[11,18–21]. In other words, the greater complexity of a functional system may weaken integration, thereby allowing diversity to be generated in different directions beyond the major defining trade-off.

Multiple complex vertebrate functional systems appear to follow a general model of diversification where a global trade-off combines with relatively weak integration to assemble high functional diversity[21–24]. This model includes an overarching trade-off defining a primary axis of trait variation, such as the axis of elongation in the mammalian skull related to

[1]Department of Fisheries, Wildlife and Conservation Sciences, Oregon State University, Corvallis, OR, USA. [2]Department of Evolution & Ecology, University of California, Davis, CA, USA. [3]Department of Biosciences, Rice University, Houston, TX, USA. [4]School of Mathematical and Natural Sciences, Arizona State University–West Valley Campus, Glendale, AZ, USA. [5]Institute of Ecology and Evolution, University of Bern, Bern, Switzerland. [6]Fish Ecology and Evolution, EAWAG, Kastanienbaum, Switzerland. [7]Department of Biological Sciences, Virginia Polytechnic Institute & State University, Blacksburg, VA, USA. [8]School of Biological Sciences, Washington State University, Pullman, WA, USA. [9]Department of Biological Sciences, Purdue University, West Lafayette, IN, USA. ✉e-mail: michael.burns@oregonstate.edu

bite force and jaw closure speed[25–27]. Considerable functional and ecological diversity evolves along this relatively integrated axis, but additional diversity also emerges independent of the major trade-off, made possible by morphological complexity and the increased evolutionary independence of traits. For instance, in carnivoran mammals, the decoupled evolution of the cranium and mandible led to lineages adapting to multiple dietary regimes while keeping clade-specific adaptations needed for sensory structures[28]. Many studies have identified an association between ecological shifts and changes in patterns and magnitudes of evolutionary integration, where ecological diversity and novel trait combinations result in varying degrees of integration and trait independence[11,22,29,30]. We explore whether this potentially general model explains the exceptional diversity of the feeding system of coral reef fishes, a highly diverse functional system that supports extensive ecological diversity[26–29].

Reef fishes are a promising candidate for this model as they are thought to show a dominant functional trade-off in their feeding mechanism. This trade-off is associated with the contrast between capturing prey using suction and biting prey attached to the substrate, the two most common feeding modes in reef fishes. Suction feeding involves the rapid expansion of the mouth and buccal cavity to generate a flow of water that pulls in prey[31–33], and it is often paired with a variable amount of forward swimming (i.e., 'ram')[34,35]. To increase suction performance, suction feeders often evolve elongate and highly kinetic oral jaw elements, including lengthened upper and lower jaw elements. Further, suction feeders often show considerable protrusion of the upper jaw, extensive expansion of the mouth cavity, and the formation of a circular mouth aperture[31,33,36–39]. Conversely, biting involves removing resources directly from the substrate[40–45]. Biters depend to varying degrees on forces exerted by the oral jaws to dislodge food from the substrate[39,46–49]. In contrast to suction feeders, biters are expected to have smaller and more robust oral jaw elements, thus increasing the force transmission, strength in resisting forces, and mechanical stability within the oral jaw lever systems[39,46–49]. This contrast in key functional attributes between skeletal mobility in suction feeding vs. strength in biting reflects the classic mechanical trade-off in skeletal linkage systems. This trade-off between the transmission of motion and force has long been thought to represent the major axis of diversity in the feeding mechanism across fishes[50,51].

Despite its importance in the fish feeding system, this trade-off cannot fully account for the astonishing trophic diversity found among coral reef fishes[52–55]. Coral reef fishes feed on virtually all reef animals, plants, and some microbial organisms[54,56–58], suggesting the potential for extensive fine-tuning of feeding morphology extending beyond the force vs. mobility trade-off. Within biters, there are lineages of herbivores that crop algal turfs[59]; detritivores that scrape detritus from the substrate[60]; and invertivores that remove sessile invertebrates[44,61,62] or that feed on structurally defended prey like urchins and mollusks[56]. Suction feeders include many general carnivores that feed on large mobile prey like other fish and cephalopods; mobile invertivores that capture invertebrates residing within the reef such as polychaetes and many crustaceans; and planktivores that feed on a diversity of organisms in the water column[63–66]. Such ecological diversity highlights the potential for the evolution of resource-specific, morphological, and functional specialization of feeding structures, resulting in enhanced feeding performance.

Coral reef fishes are a highly polyphyletic group of over 70 families, reflecting a long history of transitions on and off of reef habitats[67]. Most modern reef fishes belong to the large teleost radiation of spiny-rayed fishes, Acanthomorpha, and represent lineages that have been associated with reefs across such varied timescales as 20–150 Mya[68]. Species accumulation and ecological diversification of modern reef lineages have occurred in tandem with the ecological restructuring of reefs following the end-Cretaceous mass extinction[59,69–72]. Benthic biting lineages, in particular, have been growing continuously in number and ecological importance since at least the early Eocene[73,74]. This complex history has contributed substantially to the morphological and functional diversity of reef fishes[75,76].

To understand how reef fish feeding mechanisms have evolved to support their extensive ecological diversity, we applied phylogenetic comparative methods to morphological traits capturing the diversity of fish feeding mechanisms, measured from 110 species representing 43 major coral reef fish families. Our study aims to explore (1) whether the primary axis of variation in the feeding mechanism reflects a trade-off between suction and biting, (2) how these two feeding modes relate to morphological diversity and evolutionary patterns, and (3) whether adaptation to seven major diet categories is better explained by moving along the major trade-off, or through independent trait diversification. Our study demonstrates that diversification of the feeding apparatus across broad lineages of coral reef fishes involves the presence of the well-established mechanical trade-off between craniofacial mobility and biting performance, combined with extensive, diet-specific, trait evolution that has capitalized on the complexity and weak integration of feeding structures to support one of the most ecologically diverse assemblages of vertebrates on Earth.

## Results

### A functional morphospace for the feeding mechanism of reef fishes

First, we explored the major axes of variability in the reef fish feeding apparatus to determine if the largest axis of morphological diversity in measured traits aligns with the mechanical trade-off in musculo-skeletal morphology between suction feeding and biting. A principal components analysis (PCA) of 13 variables (Fig. 1) describing the linear dimensions of the feeding apparatus in 110 species from 43 coral reef fish families shows that the major axis of variation, PC1, largely separates species that feed by suction from species that feed by biting (Fig. 2 and Supplementary Fig. 1). The species examined showed considerable variation in oral jaw and craniofacial morphology (Fig. 2 & Supplementary Fig. 1). Taxa varied widely in the shape, size, and positioning of the oral jaw across PC1. For example, species with higher values of PC1 (e.g., triggerfish and boxfish) have small, anteriorly placed oral jaws, while species with smaller values of PC1 (e.g., lizardfish and frogfish) have large, posteriorly placed oral jaws. Species also varied considerably in head morphology across PC2. Those with lower values of PC2 (e.g., moray eel and trumpetfish) have a smaller head height and larger head length. In contrast, species with high values of PC2 (e.g., frogfish) have larger head heights and smaller head lengths.

PC1 was negatively correlated with lengths of the premaxilla, maxilla, lower jaw length, and the relative position of the palatine (palatine-maxilla joint) and lower jaw joint (quadrate-articular joint; Supplementary Table 1). This axis reflects the tendency for biters to have short oral jaws, anteriorly placed in the skull, versus the longer jaws and more posterior jaw joints of suction feeders. PC2 was correlated with head height (Supplementary Table 1), where species with higher PC2 scores had larger head heights, greater jaw opening mechanical advantage, and a more ventral position of the lower jaw joint than species with lower PC2 scores.

A plot of PC scores for each of the seven diet categories showed substantial overlap among the groups amidst some general differences (Fig. 3). Species that primarily feed by biting, including herbivore-detritivores, biting mobile invertivores, and sessile invertivores, tended to have more positive PC1 scores than species depending largely on suction to capture prey, including planktivores, suction-feeding mobile invertivores, and general carnivores. General carnivores are unique amongst the diet categories in that they represent almost all the extreme phenotypes in the morphospace and are all the species of suction feeders that tend to have more positive PC1 scores indicative of biters. Omnivores, which are almost equally split between biters and suction feeders, have more positive PC1 scores, though a few species of omnivores have negative PC1 scores. There was substantial overlap amongst all the diet groups along PC1, and diet groups tended to overlap completely along principal component 2, with no clear effect of diet on PC2 score.

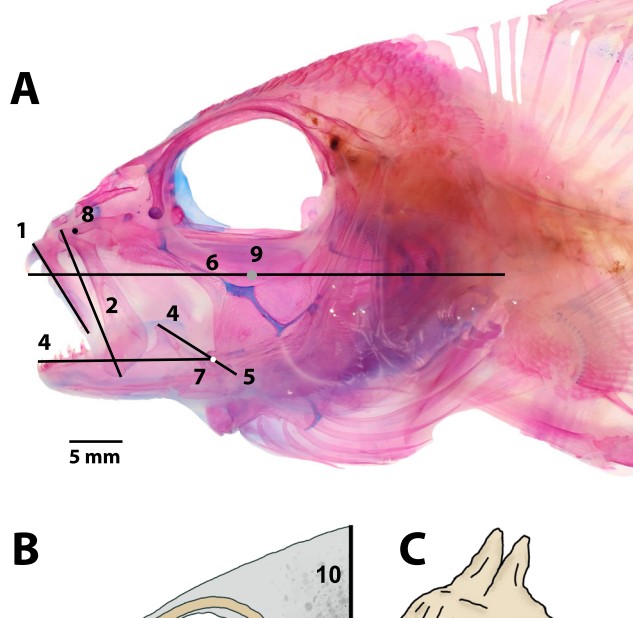

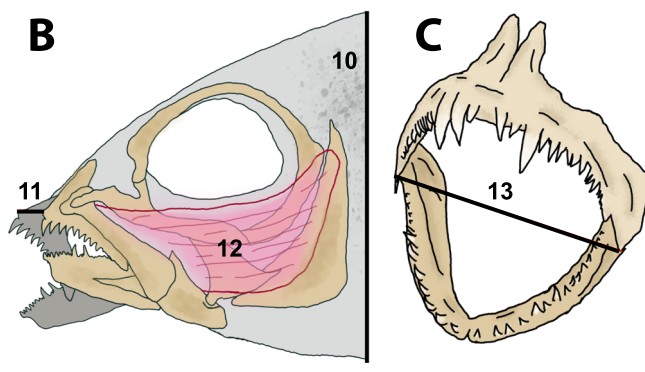

**Fig. 1 | The 13 craniofacial traits measured for each species. A** A lateral view of the head of a cleared and stained coral reef fish (*Lutjanus kasmira*) illustrating 9 of the 13 traits measured and the location of the horizontal and vertical body axis used for position measurements. (1) length of the dentigerous arm of the premaxilla, (2) length of the maxilla, (3) lower jaw length, (4) jaw closing in-lever length (used to calculate mechanical advantage) (5) jaw opening in-lever length (used to calculate mechanical advantage), (6) head length, (7) antero-posterior and dorso-ventral position of the lower jaw joint, (8) antero-posterior and dorso-ventral position of the anteriormost portion of the palatine, (9) location of the intersection of a horizontal body axis that passed through the tip of the first tooth in the premaxilla and the last vertebrate centra and a vertical axis, orthogonal to the first axis that passed through the center of the orbit of the eye. **B** Illustration of the head region showing (10) head height, (11) jaw protrusion, and (12) the adductor mandibulae. **C** Illustration of the jaws showing the (13) mouth gape measurement.

## Differences in oral jaw trait evolution between feeding modes

To further explore the morphological differences among suction feeders and biters, we used a phylogenetic MANOVA and individual phylogenetic ANOVAs that determined if and how feeding mode affects oral jaw and craniofacial morphology. The patterns within the morphospace were confirmed by the phylogenetic MANOVA, which found that suction feeders and biters differed significantly in average trophic morphology ($F = 4.01$; $p = 0.004$; Fig. 2 and Supplementary Fig. 1). Suction feeders had significantly higher jaw protrusion and longer premaxillae, maxillae, and lower jaws than biters (Fig. 4A). Suction feeders also had more posteriorly positioned lower jaw joints than biters, with the palatine and lower jaw joints also more ventral (Fig. 4A). In contrast, biters had significantly greater jaw closing mechanical advantage than suction feeders (Fig. 4A and Supplementary Table 2).

To compare which feeding mode had the most diversity in oral jaw traits, we used multivariate and univariate disparity and rate analyses. There was no significant difference in multivariate oral jaw disparity between biters and suction feeders (biters: 2.73; suction feeders: 1.99; $p = 0.2$). Biters did,

however, have significantly higher disparity in four univariate traits related to the length and positioning of the oral jaw: the lower jaw length, the antero-posterior position of the lower jaw joint, and the antero-posterior and dorso-ventral positions of the palatine (Fig. 4B and Supplementary Table 2). Patterns of oral jaw evolutionary rate were similar to disparity, with no significant difference in the rate of multivariate oral jaw evolution between biters and suction feeders (biters: 0.0027; suction feeders: 0.0025; $p = 0.6$). However, biters have higher rates of evolution across six of 13 individual traits, including in the antero-posterior and dorso-ventral position of the palatine, antero-posterior position of the lower jaw joint, lower jaw length, maxilla length, and premaxilla length. Suction feeders had a faster rate of evolution in just one trait: adductor muscle mass (Fig. 4C and Supplementary Table 3).

## Evolutionary correlation among oral jaw traits

To determine whether oral jaw and craniofacial traits were correlated, we tested whether oral jaw traits were significantly more correlated than expected under simulated uncorrelated Brownian motion. About half of the pairwise correlations between oral jaw traits (38 of 78) were significantly greater than expected if they were evolving under uncorrelated Brownian motion (Fig. 5A). Many of the significant correlations reflected intimately linked elements of the feeding apparatus, such as, positive correlations among mouth gape, premaxilla length, maxilla length, lower jaw length with the antero-posterior positionings of the lower jaw joint and palatine (Fig. 5A). However, 31 of the 38 significant evolutionary correlations were moderate or weak ($r < 0.5$) and exhibit many outliers (Fig. 5B). In all, 40 of 78 combinations were not significant, indicating that many oral jaw and craniofacial traits within coral reef fishes exhibit a high degree of independent evolution.

## Effects of diet on patterns of trait evolution

We used multi-peak Ornstein–Uhlenbeck (OU) models, estimates of disparity, and evolutionary rates to determine whether adaptation to the different diets resulted from diversification along the major trade-off or if evolution occurred independently, resulting in unique trait combinations and patterns of diversification. However, five traits, including the antero-posterior position of the palatine, the dorso-ventral and antero-posterior positions of the lower jaw joint, head height, and jaw protrusion, were based on the mean value as the trait optima could not be reliably recovered. We also used principal components analyses of these different metrics to visualize whether a variety of trait combinations, rates of evolution, and morphological variances, combined to differentiate the diets. To quantify whether different metrics of diversification adhere to the primary trade-off, we performed a linear regression of each pair of trait values to estimate the consistency of trait order across diets. A strong relationship between traits indicates similar patterns of evolution, while a weak correlation suggests mosaic evolution for different traits.

These analyses demonstrated that the trade-off between mobility and force is the main axis along which the seven diets are aligned in our dataset (Fig. 6A, B—PC1), but there is additional variation in trait combinations associated with diet diversity that occurs independently of the major trade-off (Fig. 6A, B— PC2). Optimal adductor muscle mass and the dorso-ventral position of the palatine show very little relationship with the other optimal trait values (Fig. 7A), indicating that diets can have either large or small mouth gapes with large adductor muscles and more ventrally placed jaws (Figs. 6 and 7 and Supplementary Table 3). Furthermore, even when oral jaw traits exhibit a strong relationship across most diets, some diets deviate strongly from the pattern. For instance, optimal mouth gape shows a strong relationship with optimal premaxilla length and lower jaw length, so when species in a diet category tend to have a large mouth gape, they also have a large premaxilla and lower jaw length (Fig. 7B, C). However, mobile invertivores break this pattern and have one of the smallest optimal mouth gapes and among the largest optimal premaxilla and lower jaw lengths (Fig. 7B, C). In another departure from the overall pattern, optimal mouth gape and maxilla

**Fig. 2 | Principal component analysis showing major axes of craniofacial variation in 110 species of coral reef fishes for each functional feeding mode.** Each point corresponds to a species and is colored by feeding mode. Photos show representative species illustrating morphological variation across the plot including (A) *Pristigenys serrula*, (B) *Dascyllus trimaculatus*, (C) *Acanthostracion quadricornis*, (D) *Cephalopholis baenck*, (E) *Canthigaster solandri*, (F) *Synodus saurus*, (G) *Echidna nebulosa*, and (H) *Aulostomus maculatus*.

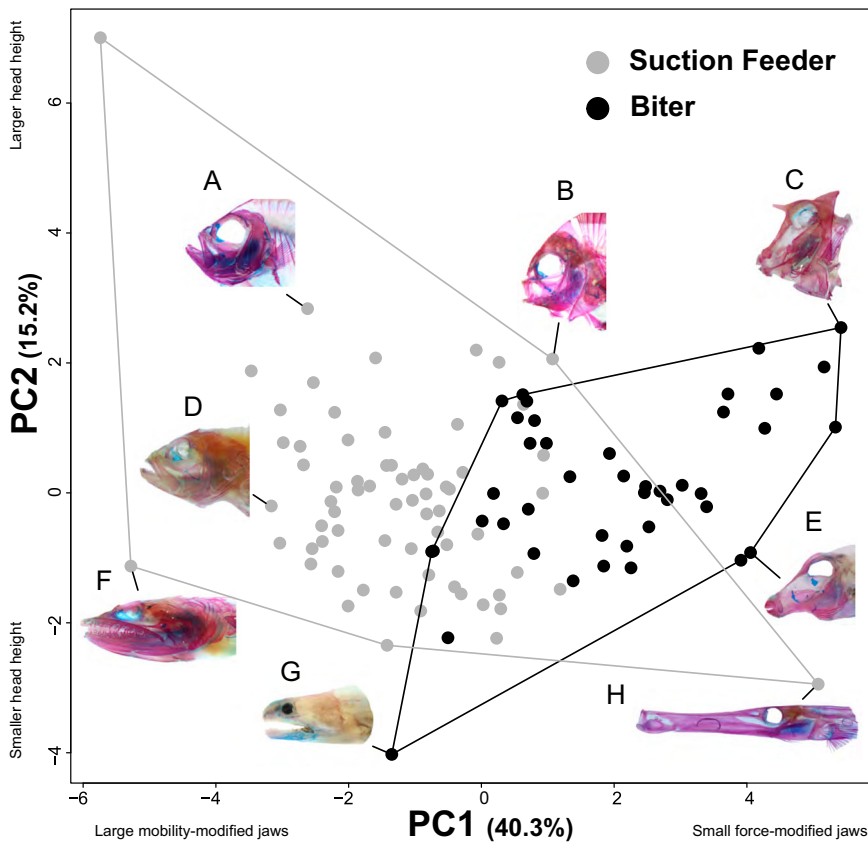

length are strongly negatively correlated with optimal closing mechanical advantage, meaning that diets with large mouth gapes and maxilla lengths tend to have small closing mechanical advantage (Fig. 7D, E). However, mobile invertivore biters deviate from this pattern, as they have some of the largest optimal mouth gapes and maxilla lengths and the second-highest optimal closing mechanical advantage (Fig. 7D, E). We found similar results using a dataset that only consisted of mean trait values (Supplementary Fig. 2). Thus, while the species in the different diets clearly diversify along the trade-off between mobility and force, further diversification in the oral jaws occurs independently of this trade-off.

Different combinations exist for rates of evolution and morphological variance. No single diet consistently showed the fastest rate or highest disparity for all 13 traits (Supplementary Figs. 3 and 4 and Supplementary Tables 4 and 5). Instead, there is high variation in the rate at which traits evolve and their contributions to generating disparity within each dietary category. Thus, our results show that the evolution of the trophic apparatus in reef fish is multi-dimensional, where diet-specific modifications exploit the complex nature of fish feeding systems to generate unique and varying combinations of trait dynamics.

## Discussion

The evolution of the feeding mechanism in coral reef fishes is dominated by a primary axis of diversification related to the mechanical trade-off between biting force and jaw mobility, which mirrors an ecological axis of feeding on attached vs. non-attached prey. Adaptation to different diets is achieved by a combination of evolution along this mechanical trade-off and adopting diverse secondary trait combinations. Thus, while the primary axis of diversity accounts for about 40% of trait variation (Fig. 2), generally weak evolutionary integration among a subset of traits allows diverse trait combinations to be associated with different

diet categories (Figs. 5 and 6). Some oral jaw traits, such as adductor muscle mass and the relative position of the palatine joint, show weak or no correlation with other traits and yet exhibit strong diet-specific patterns of evolution (Figs. 5 and 6). Such patterns suggest that the complexity of traits and evolutionary independence was crucial to adapting oral jaw elements to distinct ecological niches. The combination of this major trade-off, and the further diversification of specific morphological elements independent of this trade-off, appears to be a key component to the exceptional morphological and ecological diversity of coral reef fishes.

### Force vs. mobility underlies the primary axis of diversity

The dominant axis of variation in measured oral jaw morphology between suction feeders and biters reflects significant evolutionary correlations between gape size, the length of jaw elements, jaw closing mechanical advantage, and the position of key jaw joints in the skull. Species with large gapes, long jaw elements, low mechanical advantage, and a posteriorly placed jaw joint are exclusively taxa that feed using amplified suction. At the other extreme, species with small gapes, short jaw elements, high jaw closing mechanical advantage, and anteriorly placed jaw joints are mostly taxa that feed by directly biting prey attached to the reef. This overarching trade-off in oral jaw morphology strongly reflects the difference in functional demands of removing attached prey from the substrate versus capturing free-swimming prey from the water column. The small, robust, and more anteriorly placed jaws observed in biters more efficiently transmit bite force because of higher mechanical advantage, reflecting capacity for removing attached prey[39–45]. The longer skeletal components of the oral jaw of suction feeders create more cranial kinesis[30] and are often associated with more jaw protrusion and increased gape size, reflecting the capacity for suction mechanisms to capture elusive prey in the water column[31,33,36–39]. Thus, in our study, the well-examined mechanical trade-off in musculo-skeletal systems[24,36,39,40]

**Fig. 3 | Principal component analysis showing major axes of craniofacial variation in 110 species of coral reef fishes for each diet.** Each point is the average shape of a species, colored by diet. Large circles denote the location of the centroid for each diet.

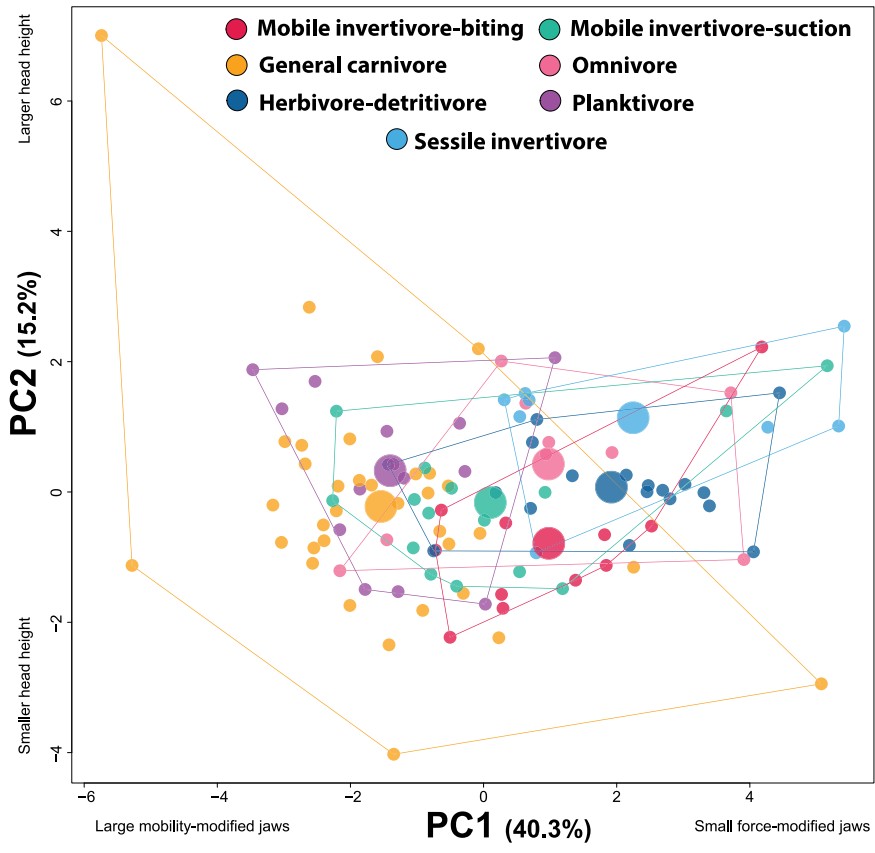

between force and mobility produces the primary axis of morphological variation in the feeding apparatus of reef fishes.

Feeding mode (suction feeding vs. biting) did not significantly affect the multivariate rate of evolution or total disparity of the trophic apparatus, but a significant effect was observed on several individual traits (Fig. 4), underscoring that trait independence has played a prominent role in diversification of feeding mechanisms. These variable patterns of evolution in oral jaw traits suggest that a system with weak integration enhances evolvability by allowing evolution to occur along and independent of the integrated axis, potentially resulting in a variety of trait values, trait disparity, and evolutionary rates[77–79]. While the impact of feeding mode on craniofacial diversification is not universal, it is clear that feeding mode impacts disparity, rates of evolution, and co-evolution among oral jaw traits.

## Diet generates a variety of trophic trait combinations

Beyond the effect of the dominant feeding mode, the trophic apparatus shows further refinement associated with different prey. Although diet categories overlap in morphospace, the distribution of the diet centroids for each group shows a strong imprint of the overarching trade-off between biting and suction feeding in that they are spread across PC1 in a sequence that reflects their reliance on either biting or suction for prey capture (Fig. 3). Diet centroids are distributed from low-to-high values on PC1, in order of generalized carnivores, planktivores, mobile invertivore suction feeders, biting mobile invertivores, omnivores, herbivore/detritivores, and sessile invertivores.

While the morphology of species in these diet categories largely follows the suction-biting axis, there are several trait-specific departures from the main axis of variation. For instance, general carnivores use ram and suction to feed on fishes, cephalopods, and large mobile crustaceans and have among the largest optimal jaw adductor muscle size across all seven diet categories. In contrast, suction-feeding planktivores are adjacent to general carnivores in morphospace, but have the smallest adductor muscle

optimum among all diets (Fig. 6). This difference in adductor muscle mass occurs without influencing other aspects of the oral jaw morphology and may reflect the higher demands associated with restraining the large captured prey by some generalized carnivores, compared to minimal such demands in planktivores[50,80,81]. Biting mobile invertivores have larger optimal oral jaw elements, including mouth gape, premaxilla, maxilla, and lower jaw length, compared to herbivores-detritivores, omnivores, and sessile invertivores, indicating diversification along the major trade-off that potentially reflects differences in prey size[80]. Despite biting mobile invertivores and sessile invertivores evolving different oral jaw sizes, both diets evolve towards larger optimal adductor muscle masses than herbivore-detritivores, indicating evolution away from the trade-off that potentially reflects increased force demands of feeding on hard invertebrates as opposed to algae[47,50]. In other words, while a mechanical trade-off for suction and biting induces an overarching axis of diversity in reef fishes, adaptation to major diet categories involves fine-tuning the feeding apparatus in a way that manifests as combinations of trophic traits independent of the major trade-off. About half of the evolutionary correlations among trophic traits are insignificant (Fig. 5), suggesting that the lack of strong integration has relaxed evolutionary constraints within the feeding apparatus. It appears that evolutionary trait independence has been instrumental in facilitating the modification of the feeding apparatus in response to diet within the overarching patterns associated with suction and biting. In addition to diverse trait combinations, we observe diversity across diets in trait disparity and rates of evolution. Our results support previous suggestions that variable levels of evolutionary independence of traits support diversification[82–85] and can increase the potential for lineages to evolve novel morphologies with unique trait combinations[11,14,15,17,18]. For instance, the avian skull is comprised of a mosaic of traits with different strengths of integration, facilitating this hyperdiverse evolutionary radiation[21,24] and allowing birds to adapt to various dietary niches[24,86]. Furthermore, variable levels of integration in aquatic snake kinematic jaw traits allowed lineages adequate biomechanical solutions to a wide range of feeding ecologies and behaviors[22].

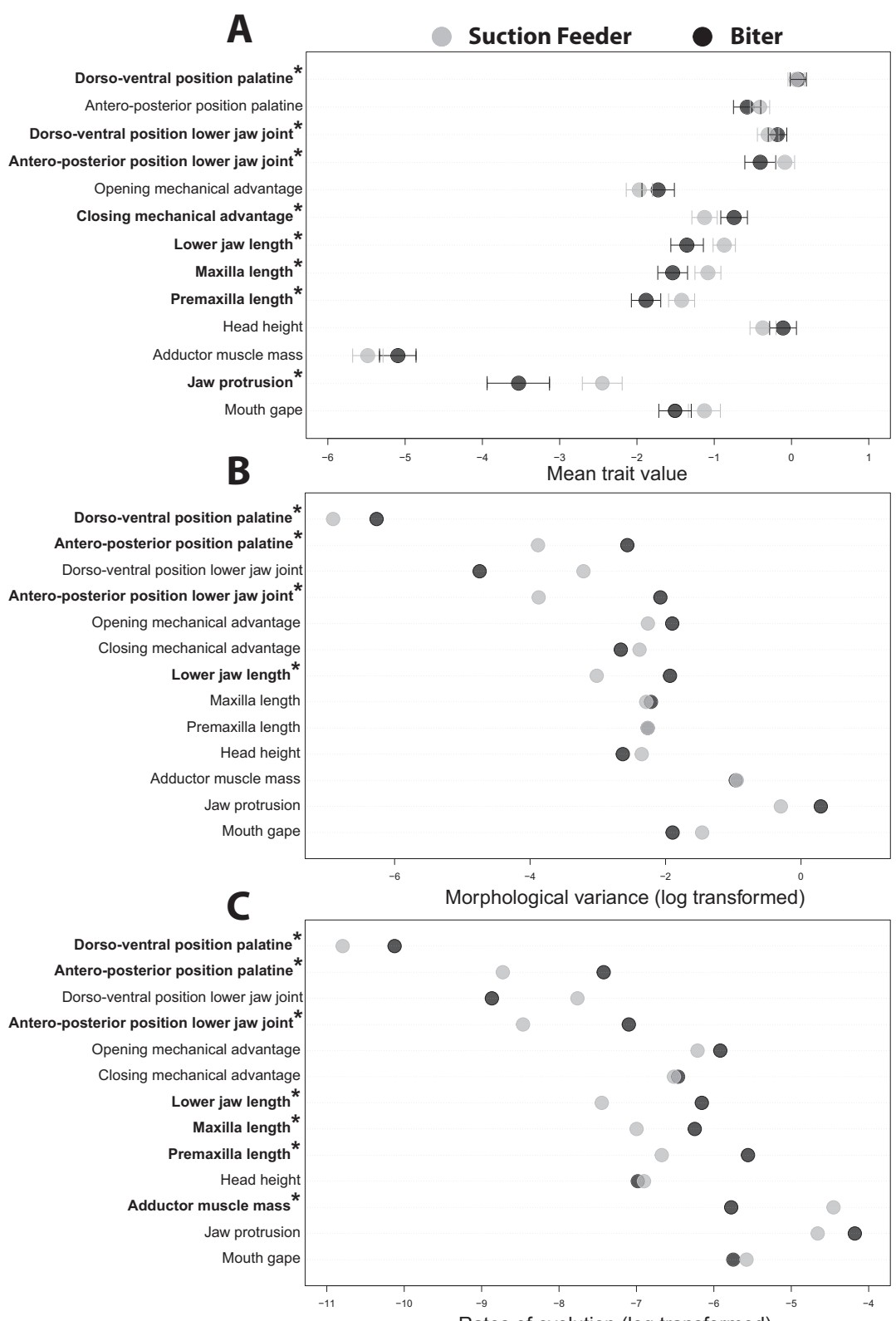

**Fig. 4 | Dot plots showing differences between biters and suction feeders for all 13 oral jaw and craniofacial traits.** Dot plots of (**A**) the mean trait value for each trait with confidence intervals around the mean, (**B**) log transformed morphological variance, and (**C**) log transformed rates of evolution. Traits that differed significantly between the functional feeding groups ($p < 0.05$) are bolded and have an asterisk.

**Fig. 5 | Heatmap of trait correlations and bivariate plots showing weak correlations between most traits. A** Heatmap of trait correlations. Blank boxes are trait correlation values that were not larger than the correlations from the Brownian motion simulation. Darker blue colors indicate a stronger positive correlation, while darker red colors indicate a stronger negative correlation. The size of the circle indicates the strength of the correlation. Letters **B–E** correspond to the bivariate plots.
**B–E** Evolutionary correlations of the independent contrasts for a sampling of oral jaw traits that show weak, but significant, correlations.

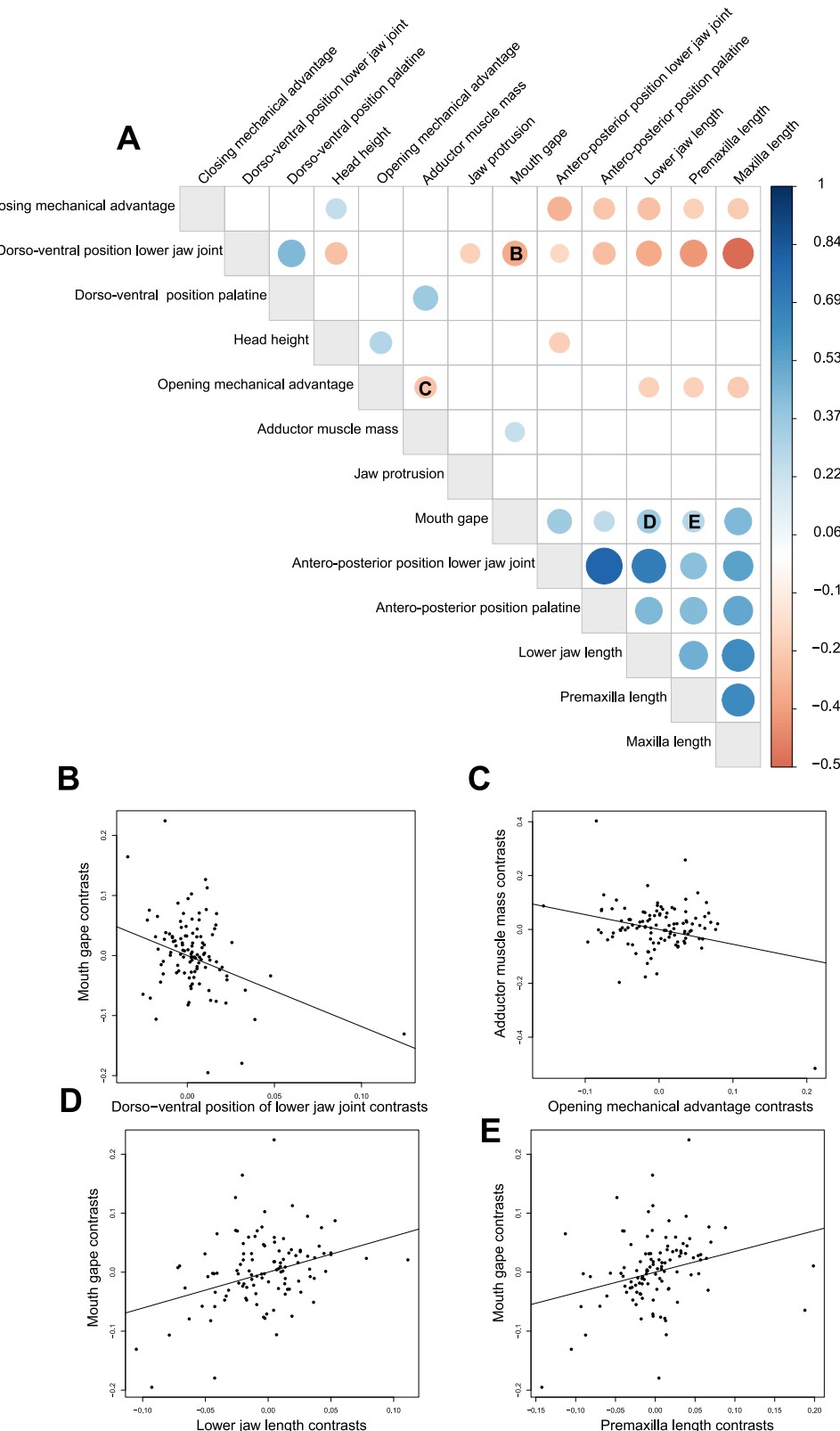

The overall pattern that emerges is that the evolution of the feeding apparatus in response to diet produces a variety of trait combinations and dynamics of trait evolution. Overall, our findings, coupled with the extensive literature on trait covariation, suggest that this pattern of weak integration between traits is a key component of complex and biologically diverse systems.

## Conclusions

The feeding apparatus of coral reef fishes is structurally and mechanically complex, phenotypically diverse, and supports exceptional ecological diversity. Like essential functional systems in other groups, understanding how they diversify is central to understanding organismal evolution and ecomorphological diversification. Our study indicates that diversification of

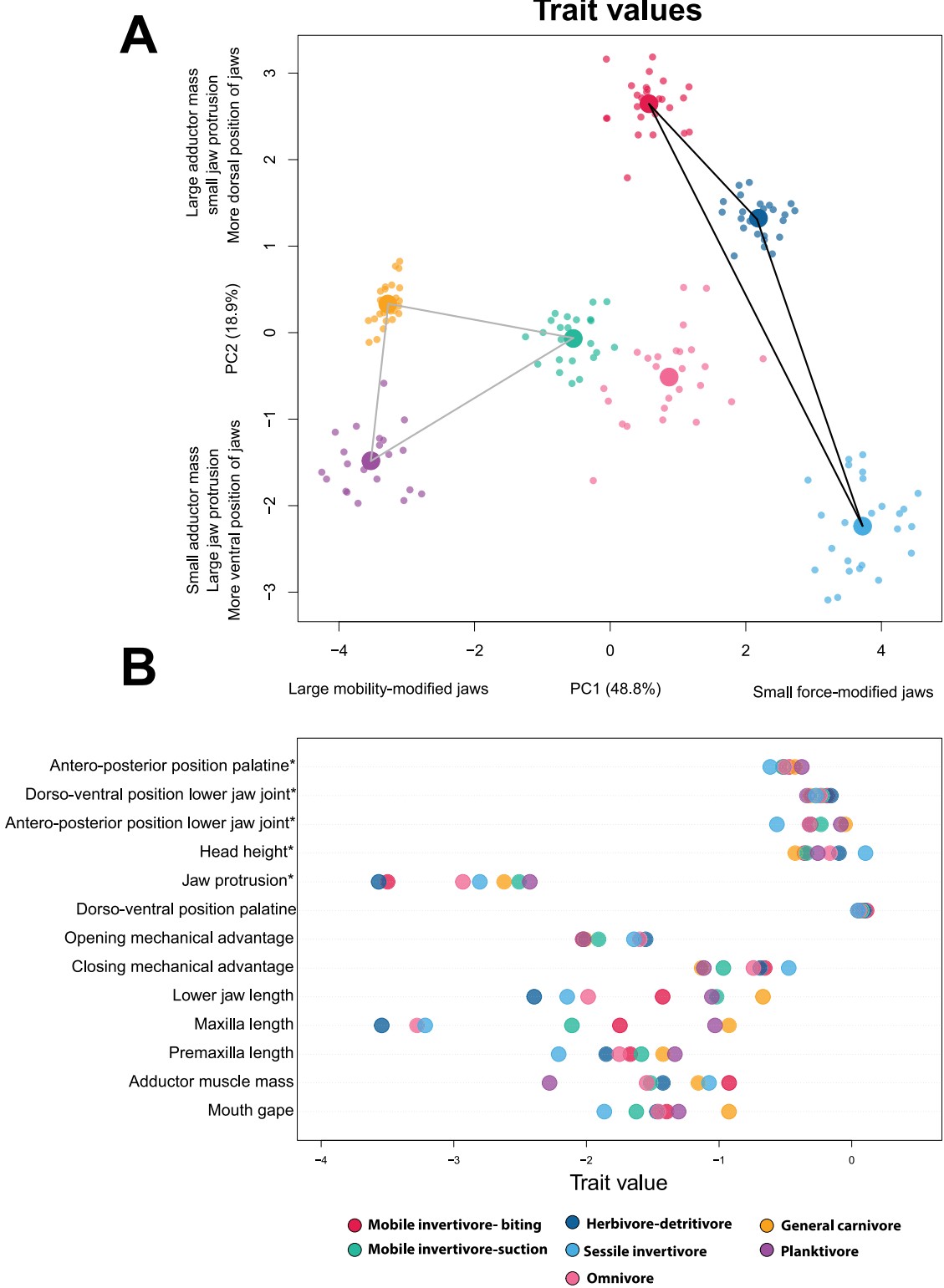

**Fig. 6 | Principal components analysis and dot plots showing differences in the optimal or mean trait values for each diet. A** Principal component analysis of optimal or mean trait values for all 13 oral jaw traits showing that, while trait combinations associated with diet are largely aligned with an overarching trade-off between jaw strength and mobility, there is variation in trait combinations beyond this trend. The black polygon represents diets that feed primarily through biting. The gray polygon represents diets that feed primarily through suction. Omnivores (pink circle) are represented by an almost equal split of biters and suction feeders. **B** Dotplot showing the variation in trait values between the diet groups for each trait. Asterisks denote traits in which the mean trait value was used instead of the optimal trait value as the multi-peak OU models could not converge.

**Fig. 7 | Heatmap of correlation coefficients and bivariate plots showing the relationships between optimal trait values for the different diets.**
**A** Heatmap of correlation coefficients from the regression analysis of diets between the optimal trait values. Darker blue colors indicate a stronger positive correlation, while darker red colors indicate a stronger negative correlation. The size of the circle indicates the strength of the correlation. Letters **B**–**E** correspond to the bivariate plots. **B**–**E** Bivariate plots of the diets for a sampling of oral jaw trait optima that exhibit deviations from the overall relationship trend despite a significant correlation.

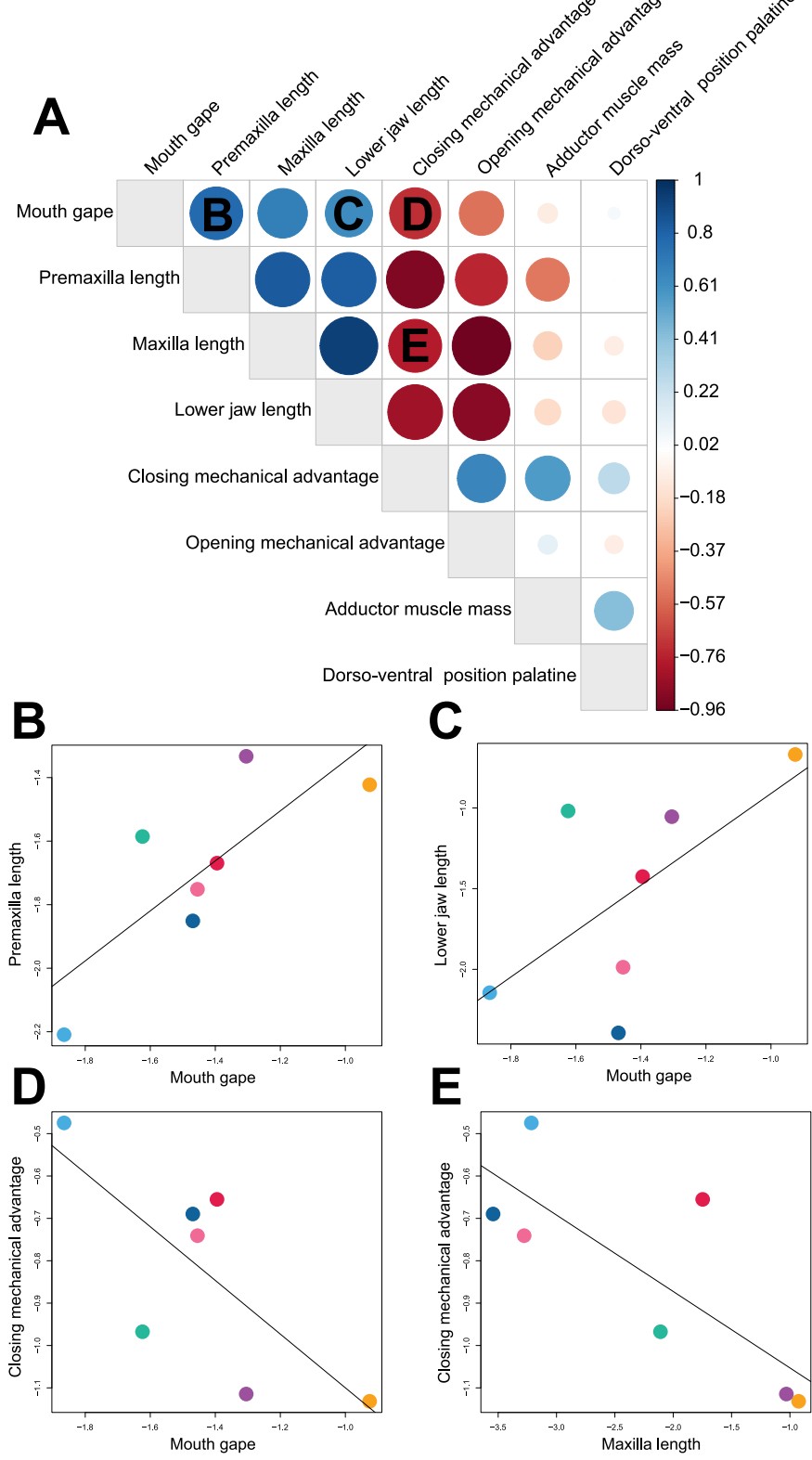

the reef fish feeding apparatus involves (1) evolution along an axis of diversity associated with a functional trade-off and (2) secondary trait modifications independent of this trade-off that result in variable trait combinations associated with ecological diversity. We expect this general model can be used to understand how the evolution of biomechanical complexity facilitates diversification in many groups of organisms across the tree of life.

## Methods
### Morphological Trait Data
We generated a morphological dataset comprising 13 traits that characterize many features of oral jaw functional morphology in reef fishes. Traits were measured on 362 specimens representing 110 species from 43 families of coral reef fishes (Supplementary Table 7). Specimens were obtained through the aquarium fish trade. All animal work in this study

was approved by the University of California, Davis Institutional Animal Care and Use Committee (protocol #22206). We have complied with all relevant ethical regulations for animal use. All live animals were of adult age and unknown sex (see Supplementary Table 7 for the species included in the study). Once in the lab, specimens were euthanized with overexposure to a solution of MS-222 and several measurements were immediately taken with dial calipers and a scale while the fish were still flexible. These measurements included mouth gape, jaw protrusion, and body mass. Fish were then fixed in 10% buffered formalin solution, after which one muscle, the adductor mandibulae, was removed by dissection and weighed. Specimens were then cleared and stained for bone and cartilage before being photographed. Photos were taken with scale bars in lateral view on a copy stand, using supports of modeling clay under slender parts of the head to position the lateral perspective of the head as close to perpendicular to the camera axis as possible. All remaining measurements were made on these photos in ImageJ (Fig. 1).

Four of the oral jaw traits were measured to characterize the position of two important joints within the head. These included the center of the joint between the anterior-most portion of the palatine and the head of the maxilla, and the antero-posterior and dorso-ventral position of the joint between the quadrate of the suspensorium and articular of the mandible (Fig. 1). To describe the position of these joints in the head, we established a 2D space centered at the intersection of a horizontal body axis that passed through the tip of the first tooth in the premaxilla and the last vertebrate centra and a vertical axis, orthogonal to the first axis, that passed through the center of the orbit. The antero-posterior and dorso-ventral positions were calculated as the x-y position away from the center of this coordinate space.

Species means were calculated for each trait and were then size-corrected using ratios. Mouth gape, jaw protrusion, head height, dentigerous premaxilla length, maxilla length, lower jaw length, antero-posterior and dorso-ventral position of the lower jaw joint, and antero-posterior and dorso-ventral position of the joint between the palatine and maxilla were divided by head length. Maximum body depth and width were divided by standard length. Adductor mandibulae mass was divided by body mass. The closing and opening in-levers lengths were divided by the jaw length, resulting in estimates of mechanical advantage. All ratios were then log-transformed to normalize the data for the different statistical analyses. Since the antero-posterior and dorso-ventral positions of the anteriormost portion of the palatine and lower jaw joint contained negative numbers, we added a standard value of 1 to each measurement to make them positive before log transforming.

## Phylogenetic Tree and Ecological Data

We pruned a large, time-calibrated phylogeny of ray-finned fishes[87] to the species in our data set. For species that were not present in the phylogeny, we used the closest related species in the tree as a proxy (see Supplementary Table 6 for substitutions). We categorized the species in the study as either biters, which includes species that use both biting and some suction, or suction feeders based on published information about their feeding habits in natural populations[73]. Biting and suction feeding is a continuum in fishes[35] and we based our classification on the primary feeding mode used to capture the dominant prey taxa in the diet. We identified 68 suction feeders and 42 biters in our dataset (see Supplementary Table 7 for classifications). We defined suction feeders as species that capture free-moving prey using some combination of suction and swimming, often referred to as 'ram', and biters as species that remove attached prey by directly biting or that must break up an armored prey to consume it, such as when triggerfishes feed on large crustaceans or urchins. We further classified each species by one of seven trophic categories modified from a previous study[88] that characterized trophic ecology across reef fishes. The trophic categories defined by Siqueira et al. (2020) were as follows: GC (general carnivore), MI (mobile invertivore), OM (omnivore), PK (planktivore), SI (sessile invertivore), and HD (herbivores-detritivores). However, we separated the mobile invertivores category into two categories, suction feeding and biting mobile invertivores, to better reflect differences in the functional challenges associated with specific prey. Suction-feeding mobile invertivores were species that fed primarily on soft-bodied invertebrates, such as polychaetes and a variety of smaller crustaceans, where the mechanical defense of the prey is minimal. Biting mobile invertivores are those species that feed primarily on armored invertebrates, like echinoderms and large crustaceans, that require extensive mechanical processing (see Supplementary Table 7 for classifications).

## Differences in oral jaw trait evolution between feeding modes and diets

We used a principal components analysis (PCA) of the 13 log-transformed trait ratios to determine the primary axes of diversity in craniofacial morphology among reef fishes. We used a PCA on the correlation matrix so that differences in measurement scale between traits would not impact our ability to visualize the variation in all traits using a PCA. We tested whether the overall construction of the oral jaw feeding apparatus (all traits) and each individual trait differed by functional and trophic group using phylogenetic MANOVA and ANOVAs, respectively. To do this, we used the procD.pgls function in *geomorph* and the pairwise function in *rrpp*[89]. Analyses were based on 10,000 permutations.

We compared morphological disparity among feeding modes and diets. We calculated oral jaw disparity for each functional group and trophic category as morphological variance using the function morphol.disparity in the R package *geomorph*[89]. We calculated the multivariate (all 13 morphological traits) morphological variance and a series of univariate variance estimates for the two feeding modes and seven diets. Each analysis was assessed for significance using 10,000 permutations.

We used state-dependent multivariate and univariate Brownian Motion models to understand whether feeding mode and diet influenced the rate of oral jaw evolution. Both multivariate and univariate Brownian Motion models were implemented in geomorph to estimate sigma, the evolutionary rate parameter[89,90]. We used feeding mode and diet as discrete traits in separate analyses. To test for significant differences in evolutionary rates between the feeding modes and diet categories, we used the permutation procedure for 10,000 iterations.

## Correlation among oral jaw traits

We calculated evolutionary correlations among the 13 oral jaw traits and tested them for significance. First, we calculated phylogenetic independent contrasts[91] for each trait using the pic function in *ape*[92] and subsequently estimated the correlation coefficient for all pairwise combinations of traits from regressions through the origin following Garland et al. [93] To determine if the correlations were greater than expected under Brownian motion evolution, we generated a null distribution of evolutionary correlations from a Brownian motion process by simulating a set of uncorrelated traits across the phylogeny using the fastBM function in *phytools*[94]. We bounded the Brownian motion simulations by each empirical trait's minimum and maximum values. We then estimated pairwise correlations of simulated traits to build a distribution of expected correlations under uncorrelated Brownian evolution. The simulations were performed 1000 times. Empirical correlations greater than the maximum correlation from the simulated dataset were deemed significant.

## Quantifying trait combinations associated with diet

We inferred the evolutionary history of the seven trophic categories using stochastic character mapping implemented in the R package *phytools*[94,95]. We generated a distribution of 1000 stochastic character maps and fit a continuous-time Markov model for the evolution of each diet using a fixed Q matrix, and we estimated the stationary distribution from the Q matrix[96,97].

We used the R package *OUwie* to fit multi-peak OU models (OUM) to diet to determine how theoretical optimal trait combinations differed between the diet groups if they were evolving under a multi-peak OU model. After estimating optimal trait values (thetas) across the entire simmap distribution, we took average theta values for each trait from across the 1000 estimates. To summarize how feeding modes and diets differed in optimal

trait space, we performed a principal components analysis on a subset of the estimated thetas for each diet and feeding mode. However, we used the mean trait value in lieu of the optima in the case that the multi-peak OU model could not converge. Since the number of observations, i.e., diets, was smaller than the number of traits, which can cause erroneous patterns during a principal component analysis, we randomly sampled 25 theta values for each diet category, giving us a total of 175 observations. Since we did not have a distribution of mean trait values to sample from, we randomly sampled 75% of the individuals in each diet category and then calculated the mean for the subsample, for a total of 25 randomly sampled means per diet. We acknowledge that traditional model-based approaches test different models of evolution to determine which model best fits the data[98]. However, we opted not to take that approach here because we were not interested in the model of evolution but rather in how the optimal trait values would combine across diet categories if they were evolving under a multi-peak OU process. To determine whether our optimal trait estimates and PCA interpretations were robust to possible model bias, we also performed a principal component analysis on mean trait values only, following the random sampling approach above. This approach allowed us to compare patterns using a dataset consisting of both optimal and mean trait values or mean trait values alone.

To determine whether each diet was associated with a unique combination of trait patterns, we performed a linear regression of the estimates of trait optima, morphological variance, and rate of evolution for each trait. This strength of the relationship between the trait combinations, rates of evolution, and morphological variances provided insight into whether the pattern of evolution is similar across traits for each diet. A strong relationship between traits in each metric indicates that the diets exhibit similar patterns of morphological evolution for each trait. In contrast, a weak relationship between traits in each metric indicates a pattern of mosaic evolution for the different traits.

## Statistics and reproducibility

All the above statistical analyses are reproducible by following the procedures in "Methods" section where the sample size and number of replicates are defined for each analysis. The Data availability and Code availability sections provide all the data and R scripts for the above analyses.

## Reporting summary

Further information on research design is available in the Nature Portfolio Reporting Summary linked to this article.

## Data availability

All data supporting the results of this study are available as Supplementary Information and archived on Figshare (https://doi.org/10.6084/m9.figshare.27245268.v1).

## Code availability

The code used to perform the analysis is archived on Zenodo (https://doi.org/10.5281/zenodo.13941776).

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

## Acknowledgements
We thank A. Adams and S. Liu for helpful comments on the figures. Two anonymous reviewers' comments greatly improved the manuscript. This research was conducted in accordance with the University of California, Davis Institutional Animal Care and Use Committee (protocol #22206). MLY was supported by a Center for Population Biology postdoctoral fellowship. ASRH was supported by the Center for Population Biology at UC Davis, the Graduate Research Fellowship under Grant No. 1650042 from the National Science Foundation, and a Dissertation Fellowship from the Ford Foundation. KTR was supported by the Graduate Research Fellowship under Grant No. 2036201 from the National Science Foundation. DKW was supported by an NSF postdoctoral fellowship (DBI-1907156). KAC was supported by an American Dissertation Fellowship from the American Association of University Women and a fellowship from the Achievement Rewards for College Scientists Foundation. Additional support and funds for the research were provided by NSF grant DEB-1061981 and by the UC Davis College of Biological Sciences to PCW.

## Author contributions
M.D.B., D.K.W., and P.C.W. designed the study. M.D.B., D.R.S., N.P., H.C., A.J.B., A.S.R., K.T.R, M.H, S.L.W. K.A.C., M.M., and D.K.W. collected the data. M.D.B. analyzed the data and wrote the manuscript. All authors contributed substantially to editing and the preparation of the manuscript.

## Competing interests
The authors declare no competing interests.
