## [Transparent Peer Review file · Communications Biology]

Complexity & Weak Integration Promote the Diversity of Reef Fish Oral Jaws.

Corresponding Author: Dr Michael Burns

This manuscript has been previously reviewed at another journal. This document only contains reviewer comments, rebuttal and decision letters for versions considered at Communications Biology.

Version 0:

Reviewer comments:

Reviewer #1

(Remarks to the Author)

In short, I think the authors did an excellent job of reframing and reworking the paper without removing any of the significance or interest to a broad array of readers. I am satisfied that my previous major comments have been addressed or clarified as needed and I think the paper is ready for publication. I included comments below that are largely semantic or just pointing out sentences that I think could be reworded. However, none of this represents anything that should hold up acceptance of the manuscript.

Minor comments:

L33-34: I would reword “functional trade-off axis” since it implies that the other axes are not functionally relevant. I think the authors could say “primary axis” since the previous sentence describes this axis well.

L48-49: This is a small stylistic comment, but I would replace the word “axis” in “improve performance on one axis”. I think it would be more informative/specific to say “function”, and it avoids overusing the word axis with slightly different meanings each time.

L55-56: I understand what this sentence is saying, but I think it could be reworded to be a bit clearer: “trade-offs may induce such strong trait integration as to limit phenotypes independent of the major trade-off axis”.

L63-64: Another very semantic point, but I don’t think genetic drift is really a force that a trait responds to. It’s more a process that a population experiences.

L90-91: I tend to think of large mouth gapes as being worse for suction feeding (smaller gape leads to higher pressure gradients and greater exerted forces).

L109-110: “mobile invertivores that capture mobile invertebrates” reads as redundant.

L145-148: I think this is backwards. In figure 2 the boxfish have high PC1 scores and the lizardfish have low PC1 scores.

L157-158: This could just say “a more ventral position of the lower jaw joint”.

L196-198: I find this sentence a bit confusing.

L281-284: I don’t think these examples add anything by being here. They feel kind of like non-sequiturs and I don’t think the fact that functional morphological tradeoffs exist needs to be proved at this point in the paper. Rather, I like how the authors worked examples in near that end of the discussion to put the results back in that broad perspective.

L289-290: I tend to think of modularity as being an emergent property of a system that has integration among some but not all of its parts. So I’m not sure what it means that evolvability is enhanced by integration in combination with modularity.

L322: Not to be pedantic, but I don’t think it’s that surprising that such a complex system is modular (not integrated across all elements).

L342-344: I think this sentence needs a bit of reworking.

Reviewer #3

(Remarks to the Author)

This manuscript has been reviewed thoroughly by two experts in the field. I read the manuscript and rebuttal with interest.

The rebuttal letter shows that all comments by the reviewers have been considered. As a functional morphologist not specialized in topics such as evolutionary integration, diversification theory, and phylogenetic comparative methods, I have no additional specific comments on the data analysis.

For a broad comparative morphological study on fish heads, I am particularly happy to see 3-dimensionality in the data set with the adductor muscle mass and mouth width measurements. Also the manipulation data of fresh specimens is valuable, and, as far as I know, exceptional for this type of analysis.

The following comments can be helpful to improve the clarity of the paper in view of the broad and non-specialized audience of Communications Biology.

(1) One reviewer comment needs more attention. Reviewer 1 wrote in comment number 2 that because the authors measured traits that are known to be important in the biting-suction axis, the data set is biased towards picking out the biting-suction axis. This comment is not resolved appropriately in my view, as the authors still write, for example that the “primary axis of variation in musculo-skeletal traits is aligned with a trade-off between mobility and force transmission” (abstract line 30), write that this trade-off is “over-arching” (line 132), that it is about determining “the largest axis of morphological diversity” (line 140), that there is “a single major axis of diversity associated with a functional trade-off” (line 345). The fact that the trade-off comes out as ‘primary’, ‘overarching’, ‘largest’, or ‘single major’ seems due to the a priori choice of variables. This seems to lean toward circular reasoning.

(2) The first paragraph is hard to follow for non-specialists. It starts with describing trade-offs projected on two axes: ‘trade-offs where changes to a component improve performance on one axis but necessarily decrease performance on a second axis’, but towards the end switches to trade-offs along a single axis: ‘the major trade-off axis’ (line 47-49). This is very confusing. In addition, in the final sentence, ‘integration’ has not been defined and introduced.

(3) Second paragraph line 60: ‘complexity of traits’: How is complexity of traits defined? Do you mean a larger number of traits? Please explain.

(4) Abstract line 36 “We suggest that morphological evolution both along and orthogonal to a major axis of variation is a widespread mechanism of diversification...”. What does it mean to be ‘orthogonal to an axis of variation’? I assume that in PCA-space this means uncorrelated traits with the traits variation on the major axis, but should readers really know that you are referring to that specific, commonly used multidimensional statistical method in your field to understand what you mean? Please describe what this means to an audience of biologists not specialized in the type of statistics common to your field.

(5) There seems to be an excessive usage of the term ‘axes of’, making the text harder to read than necessary. For every variable one can plot an axis, but usually we just write that there is variation in a variable describing a trait, not that there is variation along the axis of that trait variable. I advise the authors to go through the text and re-evaluate the necessity of ‘axes of’ in their statements. It is possible that I miss a specific meaning so in that case this comment can be ignored.

We are pleased to submit our revised manuscript entitled " Complexity & Weak Integration Promote the Diversity of Life: Reef Fish Oral Jaw Evolution. " to Communications Biology. We carefully reviewed the reviewer's comments and have addressed all of their concerns. Many thanks to the two referees for their careful review and thoughtful comments on our manuscript. We have addressed all the issues that were raised, and in so doing, we believe the manuscript has been substantially improved.

Below, we present a comprehensive point-by-point response to the reviewer's comments. We have also uploaded a manuscript file using track changes to highlight the edits and a clean copy, ensuring that all changes are clearly visible.

Reviewer #1 (Remarks to the Author):

In this manuscript the authors take a phylogenetically broad approach to address a question that is currently the focus of much attention in organismic biology; How do patterns of trait integration guide the outcomes of evolutionary radiations. To do this the authors collect phylogenetically corrected linear measurements and positional data from 110 species of coral reef fishes distributed across 43 families and use this data to create a morphospace and optimal trait space. They also estimate morphological disparity and evolutionary rates of these traits across reef fishes. They classify species based on their feeding ecology (both broadly along the suction-biting axis, and more specifically within 7 narrower niche categories) and ask whether the major axes of shape variation are predictive of these trophic categories. Finally, they model evolutionary processes to ask whether each pairwise combination of traits is integrated at a greater level than would be predicted under neutral evolution and to calculate trait optima of each narrow feeding niche.

Comment 1:

The authors use this data to make the primary claim that the evolution of complex systems proceeds through separation along a main tradeoff axis, followed by specialization along perpendicular axes in shape space. Although I think the authors could expand upon their choice of statistical methodology in several places, I think that the analysis is well done and the results are convincing. My main concern is that I think the novelty of the results are a bit overstated as this understanding of how integration and functional tradeoffs will influence evolutionary diversification is well studied across vertebrates as well as in theoretical work (particularly that the primary axis of vertebrate skull diversification is based around elongation)^{1–7}. That being said I appreciate the novelties of this study, namely the phylogenetic diversity and the use of OU models to include selection in modeling the evolution of these traits.

However, it seems like these novelties come at the cost of depth in the morphological analysis as there are relatively few measurements taken and it is restricted to linear measures. While I think that the science is well done and the manuscript is generally well written, the paper needs to be largely reframed to put it in the context of the long history of the ideas that it supports instead of claiming that it is a novel mechanism of evolutionary diversification.

Additionally, I think that several of the analytical decisions need to be further justified. I don't mean to say that the analysis is incorrect or misleading, just that I was surprised by a few choices within the methods and would like to see them explained.

We greatly appreciate your comments. We agree that there is an extensive and long history of research on phenotypic integration, how functional trade-offs may shape integration, and how varying strengths of integration can influence macroevolution. We believe our findings are completely consistent with existing theory and other empirical examples. Initially, we tried to frame our manuscript through this lens, but your comments helped us realize that our brevity caused the manuscript to lose a more nuanced interpretation. We have rewritten a portion of the introduction (Lines 53-56, 61-64, 69-83) and discussion (Lines 280-285 & 321-337) to tone down the novelty of our study, and we have now put our study in the broader context of extensive work done on evolutionary integration across vertebrates.

Comment 2:

First, I'm curious why the authors chose to use linear measurements instead of landmark based morphometrics? I would think that all of the same traits could be understood using landmarks at the endpoints of each measurement, but that this would create less constraint on how the variation in the traits is measured. I worry that by measuring traits that are known to be important in the biting-suction axis and measuring them in the one dimension that would be expected to relate to this axis, the results are biased towards picking out the biting-suction axis.

This is a very interesting point, and we appreciate how it seems like our approach might be biased towards picking out the functional axis of diversity. Indeed, we purposely selected functionally relevant traits to best capture the diversity of feeding functional morphology in coral reef fish. Below we outline why we choose to use functional traits and not landmark data.

Morphological measurements come in two broadly different forms. (1) geometric morphometric landmarks, which almost exclusively focus on describing shape and size variation, and (2) linear measurements, which can either describe shape or serve as morphometric approximations to previously understood biomechanical or functional morphological models.

There is a distinction between traditional linear morphometric- and "functionally informed" approaches (see Feilich and Lopez-Fernandez 2019 for a detailed discussion of both approaches). In short, linear morphometrics can be broken down into variables that may correlate with function or ecology without an explicit underlying functional model, or studies can be conducted using traits directly derived from functional or biomechanical modeling.

Comprehensive geometric landmark coverage provides an accurate description of shape variation. Giving functional meaning to geometric morphometric data is often challenging without knowing how shape relates to function. Landmark analyses do not isolate whichever aspects of a structure are the most associated with function. The landmarks are all treated equally. A geometric-morphometrics-based approach may mask meaningful variation by emphasizing the most variant instead of the most functionally relevant (Feilich and Lopez-Fernandez 2019).

We used functionally derived linear measurements to more closely relate the relationship between ecology and morphology in coral reef fishes. We acknowledge that it is possible that a detailed landmarking approach might discover somewhat different major axes of shape variation in reef fish and we have added this qualification in our methods descriptions. We have lost some degree of morphological variation not using geometric morphometric measurements. However,

capturing functional morphological diversity better approximates the ecologically relevant axes of diversity that we were interested in.

Kara L Feilich, Hernán López-Fernández, When Does Form Reflect Function? Acknowledging and Supporting Ecomorphological Assumptions, *Integrative and Comparative Biology*, Volume 59, Issue 2, August 2019, Pages 358–370, <https://doi.org/10.1093/icb/icz070>

Comment 3:

Secondly, I'm curious why the authors chose to do rank correlation on OU estimated optimal trait values instead of regressing the actual estimated trait values? It seems like the actual trait values would be highly relevant since it's unlikely that the distribution of these traits is as regular as rank correlation assumes them to be.

We have redone the analyses using a linear regression between traits (see Lines 495-503). The regression gives us similar results to the ranked-order correlation and the pattern of difference between traits among diets is even more pronounced as the strength of the relationship is now weaker than when using the ranked-order correlation (see Lines 227-243; Figs. 6, S3).

Comment 4:

Finally, I apologize if I am misunderstanding the second PCA analysis (of OU optima estimates), but it seems like this model is overfit. As far as I can tell there are 7 data points (diet groups) that each have 13 variables (the linear measurements) meaning that there are more predictors than observations. If I am misunderstanding the model then perhaps the description of this analysis could be clarified.

Thank you for pointing out that our low number of observations relative to variables makes for an awkward principal component analysis. To address this concern, we have tried to capitalize on the inherent variation in our data set: we randomly sampled 25 theta values for each diet category to remedy this, giving us 175 observations. Since we did not have a distribution of mean trait values to sample from, we randomly sampled 75% of the individuals in each diet category. Then, we calculated the mean for the subsample, for a total of 25 randomly sampled means per diet (Lines 441-445). We then performed the PCA on the 175 observation dataset, removing any bias that the small number of samples could cause. The results using the new PCA are the same as the previous results. We have updated the results and figures to include the new PCA.

Minor comments:

Comment 5:

L66: The authors claim that this is a general model of how complex systems diversify, but don't really explain why this particular example from reef fishes should be generalizable to other clades. I think this should either be directly addressed or the language made less general.

This is a great point. We have reworked this section to more clearly state that several complex vertebrate functional systems appear to follow a general model of diversification where a global tradeoff combines with weak integration to assemble exceptional functional diversity. We then highlight that we are interested in whether this general pattern is useful in understanding the

diversification of the feeding mechanism that underlies the exceptional ecomorphological diversity of coral reef fishes.

Comment 6:

L122-126: The tradeoff between craniofacial mobility and biting performance is the best established axis of diversification among fishes. It is also well established that there is further specialization that occurs after the diversification along this axis. So I think this sentence should be rephrased to state the things that make this study novel such as the evolutionary modeling and the phylogenetic breadth.

We agree that this axis of diversity is well-established in fishes (see lines 78-95) and that further ecological diversity likely occurs away from this axis (see lines 96-107). We agree that what makes our results unique is that we show that ecomorphological diversity across broad lineages of coral reef fishes is driven by a global tradeoff that shapes a central axis of diversity and that significant additional diversity emerges as systems take advantage of weak integration and complexity to assemble functional units with many trait combinations that meet varying ecological demands across the diversity of coral reef fishes. We have updated this sentence to highlight further that the novelty comes from the scale at which we see the pattern and mention that the mechanical tradeoff is well established.

Comment 7:

L132: I worry about redundancy in some of the 13 variables used to describe the morphology of the species. In particular the position of the lower jaw joint as and jaw length seem highly redundant since the position of the joint is measured based on an axis defined in part by the front of the head and which is parallel to the lower jaw (at least in Supplementary Figure 4).

We agree that some of the traits measured seem as though they might be highly correlated. As you suggest, lower jaw length and anteroposterior position of the lower jaw joint both capture some degree of the length of the lower jaw, as the more posterior location of the joint likely indicates a longer lower jaw length. One might expect these traits to be strongly integrated and in fact the evolutionary correlation is one of the largest with a value of 0.69. However, it is worth noting that while lower jaw length exclusively captures the size of the lower jaw, the anteroposterior position of the lower jaw joint is also related to the eye's position, which includes information on the lower jaw's position in the skull. For instance, species of *Scarus* (parrotfish) show little correlation between these two traits. *Scarus* species have some of the smallest lower jaw lengths in our dataset, which means they should also have very small values of the anteroposterior position of the lower jaw. However, *Scarus* species have anteroposterior position of the lower jaw values that are more like species with large lower jaw lengths. Furthermore, our correlation analyses show that most traits are likely not redundant, as they show no significant relationship. The traits that show a significant relationship have mainly weak to moderate strengths, indicating that even correlated traits capture slightly different axes of variation in our dataset.

Comment 8:

I also worry that the pose of the fish in the cleared and stained images will affect the positional measurements, for instance in Supp. Fig. 4 in which the mouth is wide open. Was there standardization in the position of the fishes when they were imaged?

Good point. We did not fully describe how we positioned the fish before photographing. We have added a description of the fish's standardization in Lines 330-331. Specimens were oriented left to right with the mouth fully open and extended before photographing.

Comment 9:

L169, 174-175: I don't think that it is fair to make the claim that disparity or evolutionary rates are higher across all traits. Although the authors give the caveat that the difference is not significant, it is misleading to then state that the mean is higher. The purpose of the statistical test is to determine whether differences in the means are "real" or whether there is a reasonable chance that they are a product of measurement error/partial sampling of a population. So I don't think it's fair to even mention that the means are higher because of the implications.

We agree that discussing the difference in rate and disparity is odd when it is not statistically significant, which was an oversight on our part. We have updated this section to correct that error and state that multivariate disparity and rate of oral jaw evolution did not differ significantly between biters and suction feeders.

Comment 10:

L193: Given the broad readership of the journal, I think it would be helpful to define the acronym OU here.

We have updated this section to define OU as Ornstein–Uhlenbeck models.

Comment 11:

328-330: Were the specimens wild caught or lab raised? Particularly when talking about integration I think it is highly relevant to know whether specimens were raised with different food types, as the plasticity associated with different niches would inflate the measurements of integration. I don't think this is a problem for the overall analysis, since the magnitude of plasticity would be small compared to the overall morphological disparity in the dataset, but it's still important to disclose.

We have updated line 328 to state that all specimens were wild-caught on coral reefs.

Comment 12:

L351: I think it's important to justify any data transformations, in this case the log of morphological ratios.

We have updated this sentence to state that we log-transformed the ratios to normalize the data for the different statistical analyses.

Comment 13:

L380: Why were regressions forced to have an intercept of 0? Given the non-linearity of

allometric effects and of fluid dynamics as sizes get small I wouldn't expect the same relationship between different elements to hold true as their size approaches zero. However, this choice in methodology would imply that the same linear relationship should apply across all life stages and individual sizes.

Thank you for pointing out that forcing our regressions to have an intercept of 0 is confusing based on the standard assumptions of a regression analysis. However, we regressed the independent contrasts of the traits, not the trait values, and regressions of independent contrasts need to be computed through the origin (see Garland et al. 1992 Appendix 1 for a detailed mathematical explanation of why regressions of independent contrasts need to have an intercept of zero). We have updated line 380 to "First, we calculated phylogenetic independent contrasts for each trait using the pic function in ape and subsequently estimated the correlation coefficient for all pairwise combinations of traits from regressions through the origin following Garland et al. 1992".

Theodore Garland, Paul H. Harvey, Anthony R. Ives, Procedures for the Analysis of Comparative Data Using Phylogenetically Independent Contrasts, *Systematic Biology*, Volume 41, Issue 1, March 1992, Pages 18–32, <https://doi.org/10.1093/sysbio/41.1.18>

Comment 14:

L423-426: I don't find this to be a satisfying explanation of why other models of evolution weren't considered. Why is it so important to use a multi-peak OU model even if it proves to be a worse fit for the data, especially given the fact that multiple traits did not converge?

We agree that a multi-peak OU model that is not the best fit to the data can estimate biologically unreasonable theta values, which would shape our interpretation of the PCA based on the optimal trait values. We added an analysis to ensure our theta estimates were biologically reasonable. We performed the analyses using only mean trait values and not optimal trait values (see Lines 448-452) to determine if model bias might influence our interpretation of differences in trait values among diets. We found similar results when performing a PCA on a dataset that consists of both optimal and mean trait values and a dataset consisting exclusively of mean trait values (see Lines 231-232; Fig. S2).

Comment 15:

Figure 3: The three plots should be labeled A, B, and C in accordance with the figure legend. Additionally, I think the points should have error bars and p-values should be presented in a supplemental table. I think that simplifying significance to a binary trait based on $p < 0.05$ is overly reductive and I'd like to see both the p-values of significant results and nonsignificant ones.

We have updated Figure 3 to include 95% confidence intervals around the mean trait values for each functional feeding mode. The disparity and rate analyses only give a single value, so those subplots do not contain confidence intervals. We added A, B, and C labels corresponding to each subplot. Lastly, we included a supplementary table with the p-values for each trait from each of three analyses (see Table S2)

Comment 16:

Figure 4: I like the labels in Figure 6 that show which points on the heatmap correspond to each scatterplot below. I think it would increase the clarity to include similar labels here.

Thanks for pointing this out; we agree that the addition helps strengthen the figure's clarity. We have added labels to correspond each scatterplot to the heatmap.

Comment 17:

Figure 5: It seems potentially problematic that nearly half the traits could not converge, so half the data is based on means instead of optima. This seems worth addressing somewhere in the manuscript.

In Lines 425-426, we mention that we used means instead of optima when the trait optima could not converge. We have added further clarification that five traits, including the antero-posterior position of the palatine, dorso-ventral and antero-posterior position of the lower jaw joint, head height, and jaw protrusion, were based on the mean value as the trait optima could not be reliably recovered (Lines 198-201).

1. Orkney, A., Bjarnason, A., Tronrud, B. C. & Benson, R. B. J. Patterns of skeletal integration in birds reveal that adaptation of element shapes enables coordinated evolution between anatomical modules. *Nature Ecology & Evolution* 5, 1250–1258 (2021).
2. Felice, R. N., Tobias, J. A., Pigot, A. L. & Goswami, A. Dietary niche and the evolution of cranial morphology in birds. *Proc. R. Soc. B.* 286, 20182677 (2019).
3. Vincent, S., Dang, P., Herrel, A. & Kley, N. Morphological integration and adaptation in the snake feeding system: a comparative phylogenetic study. *Journal of Evolutionary Biology* 19, 1545–1554 (2006).
4. Albertson, R. C. Morphological divergence predicts habitat partitioning in a lake malawi cichlid species complex. *American society of ichthyologists and herpetologists* 2008, 689–698 (2008).
5. Conith, A. J., Kidd, M. R., Kocher, T. D. & Albertson, R. C. Ecomorphological divergence and habitat lability in the context of robust patterns of modularity in the cichlid feeding apparatus. *BMC Evol Biol* 20, 95 (2020).
6. Felice, R. N., Randau, M. & Goswami, A. A fly in a tube: macroevolutionary expectations for integrated phenotypes. *Evolution* 72, 2580–2594 (2018).
7. Goswami, A., Smaers, J. B., Soligo, C. & Polly, P. D. The macroevolutionary consequences of phenotypic integration: from development to deep time. *Philosophical Transactions of the Royal Society B: Biological Sciences* 369, 20130254 (2014).

Reviewer #1 (Remarks on code availability):

I did not run the code myself, but looked at it to understand some of the presented analyses. All the analyses that I examined looked properly implemented.

Reviewer #2 (Remarks to the Author):

General remarks :

This study presents a 'model for the diversification of complex functional systems' (global tradeoff shapes a major axis of diversity and additional diversity emerges from different traits combinations allowed by weak integration), and suggests that it is a pervasive mechanism of diversification. It is unclear to me if other studies support this 'pervasive mechanism' hypothesis or if it is a suggestion for further explorations in other groups. Comparisons to previous studies should be expanded in the discussion to lead to the suggestion that it might be pervasive. The introduction includes necessary information. The structure of the results should match the m&m. M&m includes the necessary informations ; the measured traits should however be illustrated in the main text (not supplementary) and better described for non-fish specialists. The analyses support the conclusions and do not include major flaws.

I recommend publication with the following revisions :

Comment 1:

24- Abstract

« We show that the primary axis of variation in musculo-skeletal traits is aligned with a tradeoff between mobility and force transmission, spanning species that capture prey with suction and those that bite attached prey. »

« The dramatic trophic diversity found among reef fishes is characterized by a major tradeoff between jaws built for biting vs. Mobility »

\ These two sentences are basically the same, avoid repetitions.

We have updated the second sentence to avoid repetition.

Comment 2:

The last sentence is very long, has to be divided.

We have now split the last sentence into three to make it more readable.

Comment 3:

65- beyond the major defining tradeoff.-> tradeoff

We have changed "traedoff" to "tradeoff".

Comment 4:

90-93 – Too long sentence, split it up

We have split the sentence into two: "This contrast in key functional attributes between skeletal mobility in suction feeding vs. strength in biting reflects the classic mechanical tradeoff in skeletal linkage systems. This tradeoff is between the transmission of motion and force that has long been thought to represent the major axis of diversity in the feeding mechanism across fishes^{43,44}"

Comment 5:

107-109 - Most modern reef fishes belong to the large teleost radiation of spiny-rayed fishes, Acanthomorpha, and represent lineages that are associated with reefs since 20-150 Mya

We have changed the sentence to read, "Most modern reef fishes belong to the large teleost radiation of spiny-rayed fishes, Acanthomorpha, and represent lineages that have had associations with reefs for 20-150 Mya"

Comment 6:

118- We first ask whether the primary axis of variation \diamond explore, investigate
Can be structured like : Our study aims to explore (1) whether the primary axis.... (2)...(3)

We have restructured this section as follows: Our study aims to explore (1) whether the primary axis of variation reflects a tradeoff between suction and biting, (2) how these two feeding modes relate to morphological diversity and evolutionary patterns, and (3) whether adaptation to seven major diet categories is better explained by moving along the major tradeoff, or through independent trait diversification.

Comment 7:

137 – « upper half» and « lower half » of PC1. PC1 is the horizontal axis, how can there be a upper and lower sides...

We have changed the sentence to read "with species with lower values of PC1 (e.g., triggerfish and boxfish) having small, anteriorly placed oral jaws, while species with larger values of PC1 (e.g., lizardfish and frogfish) have large, posteriorly placed oral jaws."

Comment 8:

150 – « Species that primarily feed by biting, including herbivore-detritivores, biting mobile invertivores, and sessile invertivores, tended to have more positive PC1 scores than species depending largely on suction to capture prey, including planktivores, suction-feeding mobile invertivores, general carnivores, and omnivores. »

Centroids for mobile invertivores-suction and omnivores are positive on PC1, so say : « tended to have higher PC1 scores than... » But I wouldn't say that for the omnivores as their centroid is at the same level than mobile invertivores-biting. Even more so that later on (Fig.5) you don't classify omnivores as suction, but say that they are 'almost equal split of biters and suction feeders', be consistent.

Thank you for pointing this out. We removed omnivores from the sentence and added a new sentence (Lines 153-155) that states, "Omnivores, which are almost equally split between biters and suction feeders, have more positive PC1 scores, though a few species of omnivores have negative PC1 scores."

Comment 9:

It's interesting that all the extremes on both PC1 and PC2, all the ones that do not overlap with any other trophic categories (show unique morphologies) are general carnivores. And the suction feeders that are 'misplaced' on the biter-suction axis (among biters, on the biters' side of PC1) are also general carnivores. A word about that ? They have more extreme morphologies and

tendency to deviate from this general axis.

Thank you for pointing out the unique qualities of general carnivores. We agree that their morphospace occupation warrants more discussion in the manuscript. We have updated the results section to highlight that general carnivores are unique amongst trophic groups. Many species have unique and extreme morphologies while exhibiting phenotypes mismatching their functional feeding mode (see lines 156-161).

Comment 10:

167-179 – Figure 3, annotate them as A (trait value), B (disparity) and C (rates), and refer as such in the text.

We annotated Figure 3 with A (trait value), B (disparity), and C (rates) and referred to each annotation in the main text.

Comment 11:

175-176 – « However, biters have higher rates of evolution across nine of 13 individual traits (Fig. 3), including in the antero-posterior and dorso-ventral position of the palatine, antero-posterior position of the lower jaw joint, lower jaw length, maxilla length, and premaxilla length (Fig. 3) » That's only 6, where are the 3 others ?? On Fig.3C only 6 are also significantly higher for biters. Also refer to Fig. 3 only once at the end of this sentence.

Evolutionary correlations among oral jaw traits

Thank you for catching our mistake. We have updated the text to reflect that biters have a faster rate of evolution in six of 13 traits (Lines 184). Also, we now refer to Fig. 3 twice in the paragraph because we now highlight Fig. 3B and Fig. 3C inline.

Comment 12:

184 – « About half of the pairwise correlations, 37 of 72, between oral jaw traits were significantly greater than expected if they were evolving under uncorrelated Brownian motion (Fig. 4). »

There's an issue with the numbers. There are 78 combinations of 2 traits, not 72. I count 38 significant combinations on your Fig.4A, and 40 non significant. Please check all numbers.

Thank you very much for catching the error on our part. We have updated the text to say that "About half of the pairwise correlations, 38 of 78, between oral jaw traits were significantly greater than expected if they were evolving under uncorrelated Brownian motion (Fig. 4A)".

Comment 13:

Refer as Fig. 4 A and B in the text

We have updated lines 185-195 to refer to Fig. 4A and Fig. 4B.

Comment 14:

I don't understand why only 4 correlations are represented on Fig.4B, why those and not the others ? There is no description of the figure in the text.

We have added text highlighting that some significant correlations exhibit many outlier species that deviate from the trend line and refer to Figure 4B.

Comment 15:

203-206- no reference of Fig.5B in the text

We have added a reference to Fig 5B in lines 212 and 214.

Comment 16:

206-222 refer as Fig.6 A,B,C,D,E

We have now referenced Fig. 6 A-E in the main text (see lines 215-229).

Comment 17:

213- « However, mobile invertivores deviate from this pattern and have one of the smallest optimal mouth gapes and among the largest optimal premaxilla and lower jaw lengths (Fig. 6) »
◇ mobile invertivores-suction

Why is the relationship inverted ? If ranks are classified from lowest to highest, it should be the opposite. Mobile invertivores suction have small mouth gapes and large premaxilla and lower jaw length, as seen on Fig. 5. But on Fig. 6 high ranks of mouth gapes (which should correspond to higher values of the trait) are associated with low ranks of premaxilla and lower jaw length... It means the high values of the traits were ranked low and the low values high. Were the trait values taken as absolute values ?

Thank you for pointing out the confusion. We used a ranked order to illustrate the patterns, so a value of 1 means it is the largest value and a value of 7 is the smallest. However, we have redone this analysis using the raw trait values instead of ranked order as per reviewer 1's suggestions. The results are mostly the same, but should now be more intuitive to interpret than the ranked order figures. See lines 220-224 and Figure 6.

Comment 18:

238-239- « Some oral jaw traits, such as adductor muscle mass and the relative position of the palatine joint, show weak or no correlation with other traits and yet exhibit strong diet-specific patterns of evolution. » refer to figures as in the previous sentences

We have added a reference to the figures.

Comment 19:

Major axes of diversity is structured along a tradeoff, and this is compared to bird wings, turtle carapaces, and frog legs. This was already said in the introduction. In the discussion, the comparison to other case studies should be expanded. How is what you find similar to these

examples or others?

We have expanded to this section to discuss how our results are similar to birds and turtles in that feather length and the shape of the wing tip differ in response to aerodynamic force production tradeoffs related to flight performance in birds⁵, or how turtle carapaces exhibit a tradeoff between strength, hydrodynamic efficiency, and self-righting ability between aquatic and terrestrial clades⁶.

Comment 20:

291 - « This difference in adductor muscle mass occurs without influencing other aspects of the oral jaw morphology and may reflect the higher demands associated with restraining the large captured prey by some generalized carnivores, compared to planktivores »

We have added higher before demands in this sentence.

Comment 21:

301 – Are there studies of other taxa that found patterns of ecomorphological diversity similar to your findings that « while a mechanical tradeoff induces an overarching axis of diversity, adaptation to major diet categories involves fine-tuning the feeding apparatus in a way that manifests as combinations of trophic traits independent of the major tradeoff» Expand the comparison to previous findings, is this pattern pervasive as well ? Or other studies tend to show strong integration of trophic traits and diversification along one force-mobility axis.

You say in the abstract « We suggest that this is a pervasive mechanism of diversification in complex systems» So, are there other studies showing similar patterns ?

Thank you for pointing out that we need to better contextualize these results with other studies. We have updated Lines 320-333 to better connect how our results are shared across disparate vertebrate clades including birds and snakes.

Comment 22:

317-318- « Like essential functional systems in other groups understanding the principles of how they diversify is central to understanding organismal evolution and ecomorphological diversification » Revise the sentence.

We have revised the sentence to state, "Like essential functional systems in other groups, understanding how they diversify is central to understanding organismal evolution and ecomorphological diversification."

Comment 23:

General comment : the order of description of the methods in M&M is not the same as in results. Eg. The M&M first describes 'correlations among oral jaw traits' (376) then the PCAs (390), Results talk first about the PCAs (from 129), and correlations is after (180), same for the other methods, use the same order.

We have reordered the methods to reflect the order of results.

Comment 24:

330-335- Write what you did in chronological order

- First several measurements were taken using digital calipers on specimens
- Then the specimens were cleared and stained and lateral photographs were taken
- Linear measurements were taken in ImageJ on these photos

We have rewritten this section to put what we did in chronological order (Lines 330-338). First, several measurements were taken using digital calipers on freshly euthanized specimens. These measurements included mouth gape, jaw protrusion, body width, body depth, standard length (SL), and body mass. One muscle, the adductor mandibulae, was removed by dissection from each preserved specimen and weighed. Then, specimens were oriented left to right with the mouth fully open and extended before photographing. We took lateral photographs of cleared and stained specimens and made linear measurements in ImageJ.

Comment 25:

In supplementary S4 related to these 13 traits, it says that it illustrates 10 of the 13 traits measured, not explaining what are the 3 remaining and where they are illustrated ?

We have now illustrated the measurements in the figure.

Comment 26:

343-344. « The antero-posterior and dorso-ventral positions were calculated as the distance away from the center of this coordinate space »

I don't really understand this, maybe reformulate (a position cannot be a distance...) and add a small illustration besides S4

We have changed the sentence to read "The antero-posterior and dorso-ventral positions were calculated as the x-y position away from the center of this coordinate space."

Comment 27:

The illustration of traits measured should be in the main document, and illustrate all traits that are used in the study, including the ones taken with digital calipers. Non-fish specialists do not know how eg. Jaw protrusion, or mouth gape are measured, where on the body you take the 'body depth' measure... This illustration could also include the location of the adductor mandibulae.

We have updated the figure to include all of the measurements used in the study and added the figure to the main text.

Comment 28:

358-359 – biters include species that do both biting and suction. And suction feeders are only suction. Why the species that do both were classified as 'biter'? (and not 'suction?'), why they are not a different category ? So we could see on the PCA if they are in the middle of the axis, as a tradeoff between the two extremes ?

Biting and suction feeding are continuums in fish (see Longo et al. 2016), and almost all fish do some degree of suction when feeding. We opted to classify species based on their primary feeding mode, which for suction feeding is almost exclusively using suction to catch mobile prey. For biters, it is mostly biting to remove attached prey, though some degree of suction occurs after biting. We have updated the methods (Lines 374-376) to reflect this more nuanced explanation.

Sarah J. Longo, Matthew D. McGee, Christopher E. Oufiero, Thomas B. Waltzek, Peter C. Wainwright; Body ram, not suction, is the primary axis of suction-feeding diversity in spiny-rayed fishes. *J Exp Biol* 1 January 2016; 219 (1): 119–128.
doi: <https://doi.org/10.1242/jeb.129015>

Comment 29:

370 – separation of mobile invertivores into suction feeding and biting mobile invertivores. Why do you do that with those but not with other trophic categories? Other categories includes both suction-feeders and biters, even if some are primarily one or the other.

We mention in lines 380-387 that "...we separated the mobile invertivores category into two categories, suction feeding and biting mobile invertivores, to better reflect differences in the functional challenges associated with specific prey. Suction-feeding mobile invertivores were species that fed primarily on soft-bodied invertebrates, such as polychaetes and a variety of smaller crustaceans, where the mechanical defense of the prey is minimal. Biting mobile invertivores are those species that feed primarily on armored invertebrates, like echinoderms and large crustaceans, that require extensive mechanical processing.

Comment 30:

375 – see Table S5 for classifications

\ S6, the table S5 is the species replacements for the phylogenetic tree

We have changed this sentence to reference Table S6 instead of Table S5.

Comment 31:

388 – I'm not sure the title is really appropriate here, it's not all modelling

We changed the heading to "Tests of Discrete and Continuous Character Variation instead of "Models of Discrete and Continuous Character Variation".

Comment 32:

390-392

« We used a PCA on the correlation matrix do that differences in measurement scale between traits would not impact our ability to visualize the full trait matrix »

Confusing, formulate more clearly. How does it impact our ability to « visualize the full trait matrix » (you mean to visualize the variation in all traits using a PCA)

We have updated this sentence to state "visualize variation in all traits using a PCA" instead of "visualize the full trait matrix."

Comment 33:

397 - > We compared morphological disparity among feeding modes and diets.

We changed the sentence "We calculated morphological disparity to determine whether oral jaw diversity is influenced by feeding mode and diet" to "We compared morphological disparity among feeding modes and diets".

Comment 34:

401 – « for each ecological group » meaning the functional groups or trophic categories ?

We meant both and realize that our wording was confusing. We have updated the sentence to read "We calculated the multivariate (a matrix of all 13 morphological traits) morphological variance and a series of univariate variance estimates for functional feeding modes and diets."

Comment 35:

406- to test for significant differences in evolutionary rates between the feeding modes

We have updated the text to state "To test for significant differences in evolutionary rates between the feeding modes, we used the permutation procedure for 10,000 iterations".

Comment 36:

408 – explain what the « phylogenetic simulation procedure » There's no mention of simulating rates before.

We have removed the sentence from the manuscript.

Figures

Comment 37:

656 and 663 – « each point is the average shape of a species » No, points are positioned according to their scores on PC1 and PC2. You can say « points correspond to species and are coloured according to feeding modes. »

We have changed this to say "Each point corresponds to a species and is colored by feeding mode".

Supplementary

Comment 38:

S1 please use the same names everywhere for your traits, eg Closing mechanical advantage was jaw closing in-lever in S4 and just « closing in-lever » in your text. Check for other traits.

Thank you for pointing out that the different terms can be confusing. The closing in-lever is used to calculate closing mechanical advantage (see lines 376 and 377). To clarify this confusion, we have updated the legend of Figure S4 (now Figure 7) to state that the closing and opening in-levers are used to calculate mechanical advantage.

Comment 39:

S4 should go in main text

We have updated and added Figure S4 to the main text (see Figure 7).

Comment 40:

52 - Table S5

Check this table there are many mistakes. Seems due to misspellings :

Chilomycterus schoepfi or schoepfii ? is changed in itself

Diodon holacanthus is changed in itself

Gymnothorax thyrsoideus or thrysoideus ? is twice

Haemulon plumierii or plumieri ? is changed in itself

Pomacentrus coelestis or coelestris ? is changed in itself

Pterois russellii or russelii ? is changed in itself

Siganus guttatus or guttatus ? is change din itself

Thank you for bringing to our attention how confusing this table was. The spelling mistakes in the table under Tree Name are spelling mistakes in the Rabosky et al. 2018 phylogeny. We left those spelling mistakes in so our work would be reproducible when retrieving the phylogeny from the Fishtree R package. However, we now realize how confusing this makes the table. We have added an asterisk next to each misspelled name and added a comment to the Table to mention. We also removed the duplicate of *Gymnothorax thyrsoideus*.

Comment 41:

54- Table S6

Define the abbreviations for Diet in the legend

Thank you. We agree that including the definitions of the diet abbreviations makes the table more accessible to the reader. We have now added the diet abbreviation definitions.

Editor

Comment 1:

We would also ask you to consider adjusting the title of your manuscript. Specifically, please consider whether the proposed conclusions are broad enough for the title to be referring to “the Diversity of Life”. Please also note that we do not allow the use of a colon (:) in the title.

We have changed the title to “Complexity & Weak Integration Promote the Diversity of Reef Fish Oral Jaws” to narrow the title’s scope to that of our conclusions.

Reviewer #1 (Remarks to the Author):

In short, I think the authors did an excellent job of reframing and reworking the paper without removing any of the significance or interest to a broad array of readers. I am satisfied that my previous major comments have been addressed or clarified as needed and I think the paper is ready for publication. I included comments below that are largely semantic or just pointing out sentences that I think could be reworded. However, none of this represents anything that should hold up acceptance of the manuscript.

Minor comments:

Comment 1:

L33-34: I would reword “functional trade-off axis” since it implies that the other axes are not functionally relevant. I think the authors could say “primary axis” since the previous sentence describes this axis well.

We have changed “functional trade-off axis” to “primary trade-off axis”.

Comment 2:

L48-49: This is a small stylistic comment, but I would replace the word “axis” in “improve performance on one axis”. I think it would be more informative/specific to say “function”, and it avoids overusing the word axis with slightly different meanings each time.

In this sentence, we have replaced both instances of the word axis with function.

Comment 3:

L55-56: I understand what this sentence is saying, but I think it could be reworded to be a bit clearer: “trade-offs may induce such strong trait integration as to limit phenotypes independent of the major trade-off axis”.

We have adjusted the sentence to read, “Trade-offs may induce such strong trait correlations as to limit phenotypes independent of the major trade-off axis.”

Comment 4:

L63-64: Another very semantic point, but I don’t think genetic drift is really a force that a trait responds to. It’s more a process that a population experiences.

That is a fair point. We have removed the discussion of genetic drift from the sentence.

Comment 5:

L90-91: I tend to think of large mouth gapes as being worse for suction feeding (smaller gape leads to higher pressure gradients and greater exerted forces).

We have removed the discussion of suction feeders generally having large mouth gapes.

Comment 6:

L109-110: “mobile invertivores that capture mobile invertebrates” reads as redundant.

We have removed the second “mobile” from the sentence, so it reads “mobile invertivores that capture invertebrates residing within the reef such as polychaetes and many crustaceans.”

Comment 7:

L145-148: I think this is backwards. In figure 2 the boxfish have high PC1 scores and the lizardfish have low PC1 scores.

Thank you for catching the backward description. We have edited to say that boxfish have high PC1 scores and lizardfish have low PC1 scores.

Comment 8:

L157-158: This could just say “a more ventral position of the lower jaw joint”.

We have removed dorso-ventral from the sentence, so it now reads “a more ventral position of the lower jaw joint”.

Comment 9:

L196-198: I find this sentence a bit confusing.

We have reworked this sentence to simplify and clarify our methodology. “To determine whether oral jaw and craniofacial traits were correlated, we tested whether oral jaw traits were significantly more correlated than expected under simulated uncorrelated Brownian motion evolution.

Comment 10:

L281-284: I don’t think these examples add anything by being here. They feel kind of like non-sequiturs and I don’t think the fact that functional morphological tradeoffs exist needs to be proved at this point in the paper. Rather, I like how to authors worked examples in near that end of the discussion to put the results back in that broad perspective.

We have removed these examples.

Comment 11:

L289-290: I tend to think of modularity as being an emergent property of a system that has integration among some but not all of its parts. So I’m not sure what it means that evolvability is enhanced by integration in combination with modularity.

That’s a fair point, and we agree that modularity is strong integration among some but not all its components. We have clarified this sentence to state that weak integration allows a phenotype to evolve along an axis of least resistance without being entirely restricted to that axis, increasing evolvability and resulting in various trait values, trait disparity, and evolutionary rates.

Comment 12:

L322: Not to be pedantic, but I don't think it's that surprising that such a complex system is modular (not integrated across all elements).

That is a great point. We have removed "Surprisingly" from the sentence.

Comment 13:

L342-344: I think this sentence needs a bit of reworking.

Thank you for pointing out that the sentence was confusing. We have reworked it to highlight more strongly that understanding how functional systems diversify is essential to understanding organismal evolution and ecomorphological diversification across the tree of life.

Reviewer #3 (Remarks to the Author):

This manuscript has been reviewed thoroughly by two experts in the field. I read the manuscript and rebuttal with interest. The rebuttal letter shows that all comments by the reviewers have been considered. As a functional morphologist not specialized in topics such as evolutionary integration, diversification theory, and phylogenetic comparative methods, I have no additional specific comments on the data analysis.

For a broad comparative morphological study on fish heads, I am particularly happy to see 3-dimensionality in the data set with the adductor muscle mass and mouth width measurements. Also the manipulation data of fresh specimens is valuable, and, as far as I know, exceptional for this type of analysis.

The following comments can be helpful to improve the clarity of the paper in view of the broad and non-specialized audience of Communications Biology.

Comment 1:

(1) One reviewer comment needs more attention. Reviewer 1 wrote in comment number 2 that because the authors measured traits that are known to be important in the biting-suction axis, the data set is biased towards picking out the biting-suction axis. This comment is not resolved appropriately in my view, as the authors still write, for example that the "primary axis of variation in musculo-skeletal traits is aligned with a trade-off between mobility and force transmission" (abstract line 30), write that this trade-off is "over-arching" (line 132)", that it is about determining "the largest axis of morphological diversity" (line 140), that there is "a single major axis of diversity associated with a functional trade-off" (line 345). The fact that the trade-off comes out as 'primary', 'overarching', 'largest', or 'single major' seems due to the a priori choice of variables. This seems to lean toward circular reasoning.

This is a very interesting point. We agree that there could be some shape variation that we are not capturing since we are using only functionally relevant traits. However, we point out that while measuring functionally relevant aspects of the anatomy likely ensures that PC1 can be interpreted functionally, the specific pattern of trait correlation is definitely not readily predicted. PC1 being related to the trade-off between mobility and force transmission, and not some other mechanical phenomenon, is a novel finding. We suggest that there would be a very interesting avenue of investigation in a future study that carefully does both – measures functional traits

and uses landmark data to comprehensively capture shape. The extent to which overall shape variation in skeletal systems can be attributed to skeletal function is an important question. To address this reviewer comment, we have altered some wording to make sure it is clear that we are not claiming that our results characterize total shape variation in coral reef fishes but rather reflect the pattern of the traits we analyzed.

See changes in Lines 30, 132, 139, 222, 250, 265, 279, 341-343.

Comment 2:

(2) The first paragraph is hard to follow for non-specialists. It starts with describing trade-offs projected on two axes: 'trade-offs where changes to a component improve performance on one axis but necessarily decrease performance on a second axis', but towards the end switches to trade-offs along a single axis: 'the major trade-off axis' (line 47-49). This is very confusing. In addition, in the final sentence, 'integration' has not been defined and introduced.

Thanks for pointing out that this paragraph was confusing. We think the discussion of two axes confused our message (as pointed out by Reviewer 1), making it seem like the functional trade-off exists on two separate axes instead of being on the same functional continuum. We have tried to clarify this paragraph by removing the discussion of the axes of diversity and instead focusing on the functional trade-off and how it can limit diversification. Furthermore, we changed integration to covariation in the last sentence to make the paragraph more accessible to a broader audience.

Comment 3:

(3) Second paragraph line 60: 'complexity of traits': How is complexity of traits defined? Do you mean a larger number of traits? Please explain.

Yes, we meant a larger number of traits possibly influenced by many different environmental or genetic factors. We have clarified this section.

Comment 4:

(4) Abstract line 36 "We suggest that morphological evolution both along and orthogonal to a major axis of variation is a widespread mechanism of diversification...". What does it mean to be 'orthogonal to an axis of variation'? I assume that in PCA-space this means uncorrelated traits with the traits variation on the major axis, but should readers really know that you are referring to that specific, commonly used multidimensional statistical method in your field to understand what you mean? Please describe what this means to an audience of biologists not specialized in the type of statistics common to your field.

Thanks for pointing out that orthogonal only makes sense in a multivariate statistical framework. We have changed orthogonal to independent, to more closely connect the pattern to the process that drives it.

Comment 5:

(5) There seems to be an excessive usage of the term 'axes of', making the text harder to read than necessary. For every variable one can plot an axis, but usually we write that there is variation in a variable describing a trait, not that there is variation along the axis of that trait variable. I advise the authors to go through the text and re-evaluate the necessity of 'axes of' in their statements. It is possible that I miss a specific meaning so in that case this comment can be ignored.

We have reduced the number of times that we discuss axes of diversity and instead tried to focus on the underlying pattern or process being discussed in each instance.